# Dimensionality Reduction has Quantifiable Imperfections: Two Geometric Bounds

**Kry Yik Chau Lui**
Borealis AI
Canada
yikchau.y.lui@borealisai.com

**Gavin Weiguang Ding**
Borealis AI
Canada
gavin.ding@borealisai.com

**Ruitong Huang**
Borealis AI
Canada
ruitong.huang@borealisai.com

**Robert J. McCann**
Department of Mathematics
University of Toronto
Canada
mccann@math.toronto.edu

## Abstract

In this paper, we investigate Dimensionality reduction (DR) maps in an information retrieval setting from a quantitative topology point of view. In particular, we show that no DR maps can achieve perfect precision and perfect recall simultaneously. Thus a continuous DR map must have imperfect precision. We further prove an upper bound on the precision of Lipschitz continuous DR maps. While precision is a natural measure in an information retrieval setting, it does not measure "how" wrong the retrieved data is. We therefore propose a new measure based on Wasserstein distance that comes with similar theoretical guarantee. A key technical step in our proofs is a particular optimization problem of the $L_2$-Wasserstein distance over a constrained set of distributions. We provide a complete solution to this optimization problem, which can be of independent interest on the technical side.

## 1 Introduction

Dimensionality reduction (DR) serves as a core problem in machine learning tasks including information compression, clustering, manifold learning, feature extraction, logits and other modules in a neural network and data visualization [16, 8, 34, 19, 25]. In many machine learning applications, the data manifold is reduced to a dimension lower than its intrinsic dimension (e.g. for data visualizations, output dimension is reduced to 2 or 3; for classifications, it is the number of classes). In such cases, it is not possible to have a continuous bijective DR map (i.e. classic algebraic topology result on invariance of dimension [26]). With different motivations, many nonlinear DR maps have been proposed in the literature, such as Isomap, kernel PCA, and t-SNE, just to name a few [31, 33, 22]. A common way to compare the performances of different DR maps is to use a down stream supervised learning task as the ground truth performance measure. However, when such down stream task is unavailable, e.g. in an unsupervised learning setting as above, one would have to design a performance measure based on the particular context. In this paper, we focus on the information retrieval setting, which falls into this case. An information retrieval system extracts the features $f(x)$ from the raw data $x$ for future queries. When a new query $y_0 = f(x_0)$ is submitted, the system returns the most relevant data with similar features, i.e. all the $x$ such that $f(x)$ is close to $y_0$. For computational efficiency and storage, $f$ is usually a DR map, retaining only the most informative features. Assume that the ground truth relevant data of $x_0$ is defined as a neighbourhood $U$ of $x$ that is a ball with radius $r_U$ centered at

$x$ [1], and the system retrieves the data based on relevance in the feature space, i.e. the inverse image, $f^{-1}(V)$, of a retrieval neighbourhood $V \ni f(x_0)$. Here $V$ is the ball centered at $y_0 = f(x_0)$ with radius $r_V$ that is determined by the system. It is natural to measure the system's performance based on the discrepancy between $U$ and $f^{-1}(V)$. Many empirical measures of this discrepancy have been proposed in the literature, among which precision and recall are arguably the most popular ones [32, 23, 20, 34]. However, theoretical understandings of these measures are still very limited.

In this paper, we start with analyzing the theoretical properties of precision and recall in the information retrieval setting. Naively computing precision and recall in the discrete settings gives undesirable properties, e.g. precision always equals recall when computed by using $k$ nearest neighbors. How to measure them properly is unclear in the literature (Section 3.2). On the other hand, numerous experiments have suggested that there exists a tradeoff between the two when dimensionality reduction happens [34], yet this tradeoff still remains a conceptual mystery in theory. To theoretically understand this tradeoff, we look for continuous analogues of precision and recall, and exploit the geometric and function analytic tools that study dimensionality reduction maps [15]. The first question we ask is what property a DR map should have, so that the information retrieval system can attain zero false positive error (or false negative error) when the relevant neighbourhood $U$ and the retrieved neighbourhood $V$ are properly selected. Our analyses show the equivalence between the achievability of perfect recall (i.e. zero false negative) and the continuity of the DR map. We further prove that no DR map can achieve both perfect precision and perfect recall simultaneously. Although it may seem intuitive, to our best knowledge, this is the first theoretical guarantee in the literature of the necessity of the tradeoff between precision and recall in a dimension reduction setting.

Our main results are developed for the class of (Lipschitz) continuous DR maps. The first main result of this paper is an upper bound for the precision of a continuous DR map. We show that given a continuous DR map, its precision decays exponentially fast with respect to the number of (intrinsic) dimensions reduced. To our best knowledge, this is the first theoretical result in the literature for the decay rate of the precision of a dimensionality reduction map. The second main result is an alternative measure for the performance of a continuous DR map, called $W_2$ measure, based on $L_2$-Wasserstein distance. This new measure is more desirable as it can also detect the distance distortion between $U$ and $f^{-1}(V)$. Moreover, we show that our measure also enjoys a theoretical lower bound for continuous DR maps. Several other distance-based measures have been proposed in the literature [32, 23, 20, 34], yet all are proposed heuristically with meagre theoretical understanding. Simulation results suggest optimizing the Wasserstein measure lower bound corresponds to optimizing a weighted f-1 score (i.e. f-$\beta$ score). Thus we may optimize precision and recall without dealing with their computational difficulties in the discrete setting.

Finally, let us make some comments on the technical parts of the paper. The first key step is the Waist Inequality from the field of quantitative algebraic topology. At a high level, we need to analyse $f^{-1}(V)$, inverse image of an open ball for an arbitrary continuous map $f$. The waist inequality guarantees the existence of a 'large' fiber, which allows us to analyse $f^{-1}(V)$ and prove our first main result. We further show that in a common setting, a significant proportion of fibers are actually 'large'. For our second main result, a key step in the proof is a complete solution to the following iterated optimization problem:

$$\inf_{W:\, \mathrm{Vol}_n(W)=M} W_2(\mathbb{P}_{B_r}, \mathbb{P}_W) = \inf_{W:\, \mathrm{Vol}_n(W)=M} \inf_{\xi \in \Xi(\mathbb{P}_{B_r}, \mathbb{P}_W)} \mathbb{E}_{(a,b)\sim\xi}[\|a-b\|_2^2]^{1/2},$$

where $B_r$ is a ball with radius $r$, $\mathbb{P}_{B_r}$ ($\mathbb{P}_W$, respectively) is a uniform distribution over $B_r$ ($W$, respectively), and $W_2$ is the $L_2$-Wasserstein distance. Unlike a typical optimal transport problem where the transport function between source and target distributions is optimized, in the above problem the source distribution is also being optimized at the outer level. This becomes a difficult constrained iterated optimization problem. To address it, we borrow tools from optimal partial transport theory [9, 11]. Our proof techniques leverage the uniqueness of the solution to the optimal partial transport problem and the rotational symmetry of $B_r$ to deduce $W$.

## 1.1 Notations

We collect our notations in this section. Let $m$ be the embedding dimension, $\mathcal{M}$ be an $n$ dimensional data manifold[2] embedded in $\mathbb{R}^N$, where $N$ is the ambient dimension. $\mathcal{M}$ is typically modelled as a Riemannian manifold, so it is a metric space with a volume form. Let $m < n < N$ and $f : \mathcal{M} \subset \mathbb{R}^N \to \mathbb{R}^m$ be a DR map. The pair $(x, y)$ will be the points of interest, where $y = f(x)$. The inverse image of $y$ under the map $f$ is called fiber, denoted $f^{-1}(y)$. We say $f$ is continuous at point $x$ iff $\mathrm{osc}^f(x) = 0$, where $\mathrm{osc}^f(x) = \inf_{U;U\,\mathrm{open}}\{\mathrm{diam}(f(U)); x \in U\}$ is the oscillation for $f$ at $x \in \mathcal{M}$. We say $f$ is *one-to-one* or *injective* when its fiber, $f^{-1}(y)$ is the singleton set $\{x\}$.

We let $A \oplus \epsilon := \{x \in \mathbb{R}^N | \mathrm{d}(x, A) < \epsilon\}$ denote the $\epsilon$-neighborhood of the nonempty set $A$. In $\mathbb{R}^N$, we note the $\epsilon$-neighborhood of the nonempty set $A$ is the Minkowski sum of $A$ with $B_\epsilon^N(x)$, where the Minkowski sum between two sets $A$ and $B$ is: $A \oplus B = \{a + b | a \in A, b \in B\}$. For example, an $n$ dimension open ball with radius $r$, centered at a point $x$ can be expressed as: $B_r^n(x) = x \oplus B_r^n(0) = x \oplus r$, where the last expression is used to simplify notation. If not specified, the dimension of the ball is $n$. We also use $B_r$ to denote the ball with radius $r$ when its center is irrelevant. Similarly, $S_r^n$ denotes $n$-dimensional sphere in $\mathbb{R}^{n+1}$ with radius $r$. Let $\mathrm{Vol}_n$ denote $n$-dimensional volume.[3] When the intrinsic dimension of $A$ is greater than $n$, we set $\mathrm{Vol}_n(A) = \infty$. Throughout the rest of the paper, we use $U$ to denote $B_{r_U}(x)$ a ball with radius $r_U$ centered at $x$ and $V = B_{r_V}(y)$ a ball with radius $r_V$ centered at $y$. These are metric balls in a metric space. For example, they are geodesic balls in a Riemannian manifold, whenever they are well defined. In Euclidean spaces, $U$ is a Euclidean ball with $L_2$ norm. By $T_\#(\mu) = \nu$, we mean a map $T$ pushes forward a measure $\mu$ to $\nu$, i.e. $\nu(B) = \mu(T^{-1}(B))$ for any Borel set $B$. We say a measure $\mu$ is dominated by another measure $\nu$, if for every measurable set $A$, $\mu(A) \leq \nu(A)$.

## 2 Precision and recall

We present the definitions of precision and recall in a continuous setting in this section. We then prove the equivalence between perfect recall and the continuity, followed by a theorem on the necessary tradeoff between the perfect recall and the perfect precision for a dimension reduction information retrieval system. The main result of this section is a theoretical upper bound for the precision of a continuous DR map.

### 2.1 Precision and recall

While precision and recall are commonly defined based on finite counts in practice, when analysing DR maps between spaces, it is natural to extend their definitions in a continuous setting as follows.

**Definition 1** (Precision and Recall). *Let $f$ be a continuous DR map. Fix $(x, y = f(x))$, $r_U > 0$ and $r_V > 0$, let $U = B_{r_U}(x) \subset \mathbb{R}^N$ and $V = B_{r_V}^m(y) \subset \mathbb{R}^m$ be the balls with radius $r_U$ and $r_V$ respectively. The **precision** and **recall** of $f$ at $U$ and $V$ are defined as:*

$$Precision^f(U, V) = \frac{Vol_n(f^{-1}(V) \cap U)}{Vol_n(f^{-1}(V))}; \qquad Recall^f(U, V) = \frac{Vol_n(f^{-1}(V) \cap U)}{Vol_n(U)}.$$

*We say $f$ achieves **perfect precision** at $x$ if for every $r_U$, there exists $r_V$ such that $Precision^f(U, V) = 1$. Also, $f$ achieves **perfect recall** at $x$ if for every $r_V$, there exists $r_U$ such that $Recall^f(U, V) = 1$. Finally, we say $f$ achieves **perfect precision** (**perfect recall**, respectively) in an open set $W$, if $f$ achieves perfect precision (perfect recall, respectively) at $w$ for any $w \in W$.*

Note that perfect precision requires $f^{-1}(V) \subset U$ except a measure zero set. Similarly, perfect recall requires $U \subset f^{-1}(V)$ except a measure zero set. Figure 1 illustrates the precision and recall defined above. To measure the performance of the information retrieval system, we would like to understand how different $f^{-1}(V)$ is from the ideal response $U = B_{r_U}(x)$. Precision and recall provides two meaningful measures for this difference based on their volumes. Note that $f$ achieves

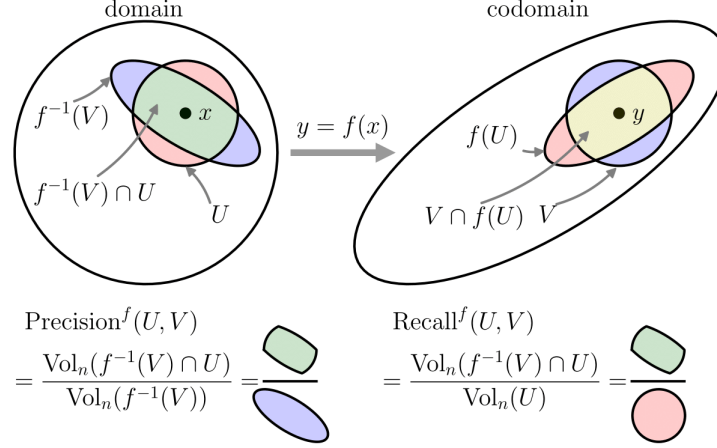

Figure 1: Illustration of precision and recall.

perfect precision at $x$ implies that no matter how small the relevant radius $r_U$ is for the image, the system would be able to achieve zero false positive by picking proper $r_V$. Similarly perfect recall at $x$ implies no matter how small $r_V$ is, the system would not miss the most relevant images around $x$.

In fact, the definitions of perfect precision and perfect recall are closely related to continuity and injectivity of a function $f$. Here we only present an informal statement. Rigorous statements are given in the Appendix B.

**Proposition 1.** *Perfect recall is equivalent to continuity. If $f$ is continuous, then perfect precision is equivalent to injectivity.*

The next result shows that no DR map $f$, continuous or not, can achieve perfect recall and perfect precision simultaneously - a widely observed but unproved phenomenon in practice. In other words, it rigorously justifies the intuition that perfectly maintaining the local neighbourhood structure is impossible for a DR map.

**Theorem 1** (Precision and Recall Tradeoff). *Let $n > m$, $\mathcal{M} \subset \mathbb{R}^N$ be a Riemannian $n$-dimensional submanifold. Then for any (dimensionality reduction) map $f : \mathcal{M} \to \mathbb{R}^m$ and any open set $W \subset \mathcal{M}$, $f$ cannot achieve both perfect precision and perfect recall on $W$.*

## 2.2 Upper bound for the precision of a continuous DR map

In this section, we provide a quantitative analysis for the imperfection of $f$. In particular, we prove an upper bound for the precision of a continuous DR map $f$ (thus $f$ achieves perfect recall). For simplicity, we assume the domain of $f$ is an $n$-ball with radius $R$ embedded in $\mathbb{R}^N$, denoted by $B_R^n$. Our main tool is the Waist Inequality [29, 1] in quantitative topology. See Appendix A for an exact statement.

Intuitively, the Waist Inequality guarantees the existence of $y \in \mathbb{R}^m$ such that $f^{-1}(y)$ is a 'large' fiber. If $f$ is also $L$-Lipschitz, then for $p$ in a small neighbourhood $V$ of $y$, $f^{-1}(p)$ is also a 'large' fiber, thus $f^{-1}(V)$ has a positive volume in $\mathcal{M}$. Exploiting the lower bound for $\mathrm{Vol}_n\left(f^{-1}(V)\right)$ leads to our upper bound in Theorem 2 on the precision of $f$, $\mathrm{Precision}^f(U, V)$. A rigorous proof is given in the appendix Appendix C.

**Theorem 2** (Precision Upper Bound, Worst Case). *Assume $n > m$, and that $f : B_R^n \to \mathbb{R}^m$ is a continuous map with Lipschitz constant $L$. Let $r_U$ and $r_V > 0$ be fixed. Denote*

$$D(n, m) = \frac{\Gamma(\frac{n-m}{2} + 1)\Gamma(\frac{m}{2} + 1)}{\Gamma(\frac{n}{2} + 1)} . \tag{1}$$

*Then there exists $y \in \mathbb{R}^m$ such that for any $x \in f^{-1}(y)$, we have:*

$$\mathrm{Precision}^f(U, V) \leq D(n, m) \left(\frac{r_U}{R}\right)^{n-m} \frac{r_U^m}{p^m(r_V/L)} \tag{2}$$

where $p^m(r)$ is $r^m (1 + o(1))$, i.e. $\lim_{r \to 0} \dfrac{p^m(r)}{r^m} = 1$.

**Remark 1.** *Key to the bound is the Waist Inequality. As such, upper bounds on precision for other spaces (i.e. cube, see Klartag [17] ) can be established, provided there is a Waist Inequality for the space. The Euclidean norm setting can also be extended to arbitrary norms, exploiting convex geometry (i.e. Akopyan and Karasev [2]). Rigorous proofs are given in the appendix C.*

**Remark 2.** *With $m$ fixed as a constant, note that $D(n,m)$ decays asymptotically at a rate of $(1/n)^{m/2}$. Also note that $r_U < R$ implies $\left(\frac{r_U}{R}\right)^{n-m}$ decays exponentially. Typically, $L$ can grow at a rate of $\sqrt{n}$. Moreover, while $p^m(r)$'s behaviour is given asymptotically, it is independent of $n$. Thus the upper bound decay is dominated by the exponential rate of $n - m$. For fixed $n, m$, this upper bound can be trivial when $r_U \gg r_V$. However, this rarely happens in practice in the information retrieval setting. Note that the number of relevant items, which is indexed by $r_U$, is often smaller than the number of retrieved items, that depends on $r_V$, while they are both much smaller than number of total items, indexed by $R$.*

*We note however that this bound depends on the intrinsic dimension $n$. When $n \ll N$ and the ambient dimension $N$ is used in place, the upper bound could be misleading in practice as it is much smaller than it should be. To estimate this bound in practice, a good estimate on intrinsic dimension [13] is needed, which is an active topic in the field and beyond the scope of this paper.*

Theorem 2 guarantees the existence of a particular point $y \in \mathbb{R}^m$ where the precision of $f$ on its neighbourhood is small. It is natural to ask if this is also true in an average sense for every $y$. In other words, we know an information retrieval system based on DR maps always has a blindspot, but is this blindspot behaviour a typical case? In general, when $m > 1$, this is false, due to a recent counter-example constructed by Alpert and Guth [3]. However, our next result shows that for a large number of continuous DR maps in the field, such upper bound still holds with high probability.

**Theorem 3** (Precision Upper Bound, Average Case). *Assume $n > m$ and $B_R^n$ is equiped with uniform probability distribution. Consider the following cases:*

- **case 1:** $m = 1$ and $f : B_R^n \to \mathbb{R}^m$ is $L$ Lipschitz continuous, or

- **case 2:** $f : B_R^n \to \mathbb{R}^m$ is a $k$-layer feedforward neural network map with Lipschitz constant $L$, with surjective linear maps in each layer.

*Let $0 < \delta^2 < R^2 - r_U^2$, $r_U, r_V > 0$ be fixed, then with probability at least $q_1$ for case 1 or $q_2$ for case 2, it holds that*

$$Precision^f(U, V) \leq D(n, m) \left( \frac{r_U}{\sqrt{r_U^2 + \delta^2}} \right)^{n-m} \frac{r_U^m}{p^m(r_V/L)}, \tag{3}$$

*where*

$$q_1 = \frac{\frac{1}{2\pi R} \int_{B_\Re^m} Vol_{n-m+1} Proj_1^{-1}(t) dt}{Vol_n(B_R^n)} , \quad q_2 = \frac{\int_{B_\Re^m} Vol_{n-m} Proj_2^{-1}(t) dt}{Vol_n(B_R^n)} ,$$

*$\Re = \sqrt{R^2 - r_U^2 - \delta^2}$, $Proj_1 : S_R^{n+1} \to \mathbb{R}^m$ and $Proj_2 : B_R^n \to \mathbb{R}^m$ are arbitrary surjective linear maps. Furthermore,*

$$\lim_{\frac{r_U^2 + \delta^2}{R^2} \to 0} q_1 = 1 \quad \lim_{\frac{r_U^2 + \delta^2}{R^2} \to 0} q_2 = 1.$$

See Appendix D for an explicit characterization of $Proj_1^{-1}(t)$ and $Proj_2^{-1}(t)$. Theorem 2 and Theorem 3 together suggest that practioners should be cautious in applying and interpreting DR maps. One important application of DR maps is in data visualization. Among the many algorithms, t-SNE's empirical success made it the de facto standard. While [5] shows t-SNE can recover inter-cluster structure in some provable settings, the resulted intra-cluster embedding will very likely be subject to the constraints given in our work[4]. For example, recall within a cluster will be good, but the intra-cluster precision won't be. In more general cases and/or when perplexity is too small, t-SNE

can create artificial clusters, separating neighboring datapoints. The resulted visualization embedding may enjoy higher precision, but its recall suffers. The interested readers are referred to Appendix G.1 for more experimental illustrations. Our work thus sheds light on the inherent tradeoffs in any visualization embedding. It also suggests the companion of a reliability measure to any data visualization for exploratory data analysis, which measures how a low dimensional visualization represents the true underlying high dimensional neighborhood structure.[5]

## 3 Wasserstein measure

Intuitively we would like to measure how different the original neighbourhood $U$ of $x$ is from the retrieved neighbourhood $f^{-1}(V)$ when using the neighbourhood of $f(x)$ in $\mathbb{R}^m$. Precision and Recall in Section 2.1 provide a semantically meaningful way for this purpose and we gave a non-trivial upper bound for precision when the feature extraction is a continuous DR map. However, precision and recall are purely volume-based measures. It would be more desirable if the measure could also reflect the information about the distance distortions between $U$ and $f^{-1}(V)$. In this section, we propose an alternative measure to reflect such information based on the $L_2$-Wasserstein distance. Efficient algorithms for computing the empirical Wasserstein distance exists in the literature [4]. Unlike the measure proposed in Venna et al. [34], our measure also enjoys a theoretical guarantee similar to Theorem 2, which provides a non-trivial characterization for the imperfection of dimension reduction information retrieval.

Let $\mathbb{P}_U$ ($\mathbb{P}_{f^{-1}(V)}$, respectively) denote the uniform probability distribution over $U$ ($f^{-1}(V)$, respectively), and $\Xi(\mathbb{P}_U, \mathbb{P}_{f^{-1}(V)})$ be the set of all the joint distribution over $B_R^n \times B_R^n$, whose marginal distributions are $\mathbb{P}_U$ over the first $B_R^n$ and $\mathbb{P}_{f^{-1}(V)}$ over the second $B_R^n$. We propose to measure the difference between $U$ and $f^{-1}(V)$ by the $L_2$-Wasserstein distance between $\mathbb{P}_U$ and $\mathbb{P}_{f^{-1}(V)}$:

$$W_2(\mathbb{P}_U, \mathbb{P}_{f^{-1}(V)}) = \inf_{\xi \in \Xi(\mathbb{P}_U, \mathbb{P}_{f^{-1}(V)})} \mathbb{E}_{(a,b)\sim\xi}[\|a - b\|_2^2]^{1/2}.$$

In practice, it is reasonable to assume that $\mathrm{Vol}_n(U)$ is small in most retrieval systems. In such cases, low $W_2(P_U, P_{f^{-1}(V)})$ cost is closely related to high precision retrieval. To see that, when $\mathrm{Vol}_n(U)$ is small, achieving high precision retrieval requires small $\mathrm{Vol}_n(f^{-1}(V))$, which is a precise quantitative way of saying $f$ being roughly injective. Moreover, as seen in Section 2.1, $f$ being roughly injective $\approx f$ giving high precision retrieval. As a result, we can expect high precision retrieval performance when optimizing $W_2(P_U, P_{f^{-1}(V)})$ measure. Such relation is also empirically confirmed in the simulation in Section 3.2.

Besides its computational benefits, for a continuous DR map $f$, the following theorem provides a lower bound on $W_2(\mathbb{P}_U, \mathbb{P}_{f^{-1}(V)})$ with a similar flavour to the precision upper bound in Theorem 1.

**Theorem 4** (Wasserstein Measure Lower Bound). *Let $n > m$, $f : B_R^n \to \mathbb{R}^m$ be a L-Lipschitz continuous map, where $R$ is the radius of the ball $B_R^n$. There exists $y \in \mathbb{R}^m$ such that for any $x \in f^{-1}(y)$, any $r_U > 0$ such that $B_{r_U}^n(x) \subset B_R^n$, and any $r_V > 0$ such that $r \geq r_U$, we have:*

$$W_2^2(\mathbb{P}_U, \mathbb{P}_{f^{-1}(V)}) \geq \frac{n}{n+2}(r - r_U)^2$$

*where $r = \left(\frac{\Gamma(\frac{n}{2}+1)}{\Gamma(\frac{n-m}{2}+1)\Gamma(\frac{m}{2}+1)}\right)^{\frac{1}{n}} R^{\frac{n-m}{n}}(p^m(r_V/L))^{\frac{1}{n}}$. In particular, as $n \to \infty$,*

$$W_2^2(\mathbb{P}_U, \mathbb{P}_{f^{-1}(V)}) = \Omega\left((R - r_U)^2\right).$$

We sketch the proof here. A complete proof can be found in Appendix E. The proof starts with a lower bound of $\mathrm{Vol}_n\left(f^{-1}(V)\right)$ by the topologically flavored waist inequality (Equation (6)). Heuristically $\mathrm{Vol}_n(f^{-1}(V))$ is much larger than $\mathrm{Vol}_n(U)$ when $n \gg m$ and $R \gg r_U$. The main component of the proof is to establish an explicit lower bound for $W_2(\mathbb{P}_U, \mathbb{P}_W)$ over all possible $W$ of a fixed volume $\mathcal{V}$, [6] where $U$ is a ball with radius $r_U$, as shown in Theorem 5. In particular, we prove that the shape

of optimal $W^*$ must be rotationally invariant, thus $W^*$ must be a union of spheres. This is achieved by levering the uniqueness of the solution to the optimal partial transport problem [9, 11]. We then prove that the optimal solution for $W$ is the ball that has a common center with $U$.

**Theorem 5.** *Let $U = B_{r_U}$ and $\mathcal{V} \geq Vol(U)$. Then*

$$\inf_{W:\, Vol_n(W) \geq \mathcal{V}} W_2(\mathbb{P}_U, \mathbb{P}_W) = \inf_{W:\, Vol_n(W) = \mathcal{V}} W_2(\mathbb{P}_U, \mathbb{P}_W) = W_2(\mathbb{P}_U, \mathbb{P}_{B_{r_\mathcal{V}}}),$$

*where $B_{r_\mathcal{V}}$ is an $r_\mathcal{V}$ ball with the same center with $U$ such that $Vol_n(B_{r_\mathcal{V}}) = \mathcal{V}$. Moreover, $T(x) = \frac{r_U}{r_\mathcal{V}} x$, for $x \in B_{r_\mathcal{V}}$ is the optimal transport map (up to a measure zero set), so that*

$$W_2(\mathbb{P}_U, \mathbb{P}_{B_{r_\mathcal{V}}}) = \int_{B_{r_\mathcal{V}}} |x - T(x)|^2 \, d\mathbb{P}_{B_{r_\mathcal{V}}}(x).$$

*Complementarily, when $0 < \mathcal{V} < Vol_n(U)$, the infimum $\inf_{W:\, Vol_n(W)=\mathcal{V}} W_2(\mathbb{P}_U, \mathbb{P}_W) = 0$, is not attained by any set. On the other hand, $\inf_{W:\, Vol_n(W) \geq \mathcal{V}} W_2(\mathbb{P}_U, \mathbb{P}_W) = 0$ by taking $W = U$.*

**Remark 3.** *Our lower bound in Theorem 4 is (asymptotically) tight. Note that by Theorem 4, $W_2^2(\mathbb{P}_U, \mathbb{P}_{f^{-1}(V)})$ has a (maximum) lower bound of scale $(R - r_U)^2$. On the other hand, by Theorem 5, $W_2^2(\mathbb{P}_U, \mathbb{P}_{f^{-1}(V)}) \leq W_2^2(\mathbb{P}_U, \mathbb{P}_{B_R^n}) = \Omega((R - r_U)^2)$, where the equality is by standard algebraic calculations.*

## 3.1 Iso-Wasserstein inequality

We believe Theorem 5 is of independent interest itself, as it has the same flavor as the isoperimetric inequality (See Appendix A for an exact statement.) which arguably is the most important inequality in metric geometry. In fact, the first statement of Theorem 5 can be restated as the following inequality:

**Theorem 6** (Iso-Wasserstein Inequality). *Let $B_{r_1}, B_{r_2} \subset B_R^n$ be two concentric $n$ balls with radii $r_1 \leq r_2$ centered at the origin. For all measurable $A \subset B_R^n$ with $Vol_n(A) = Vol_n(B_{r_2})$, we have*

$$W_2(\mathbb{P}(A), \mathbb{P}(B_{r_1})) \geq W_2(\mathbb{P}(B_{r_2}), \mathbb{P}(B_{r_1}))$$

*where $\mathbb{P}(S)$ denotes a uniform probability distribution on $S$, i.e. $\mathbb{P}(S)$ has density $\frac{1}{Vol_n(S)}$.*

Recall that an isoperimetric inequality in Euclidean space roughly says balls have the least perimeter among all equal volume sets. Theorem 6 acts as a transportation cousin of the isoperimetric inequality. While the isoperimetric inequality compares $n - 1$ volume between two sets, the iso-Wasserstein inequality compares their Wasserstein distances to a small ball. The extrema in both inequalities are attained by Euclidean balls.

## 3.2 Simulations

In this section, we demonstrate on a synthetic dataset that our lower bound in Theorem 4 can be a reasonable guidance for selecting the retrieval neighborhood radius $r_V$, which emphasizes on high precision. The simulation environment is to compute the optimal $r_V$ by minimizing the lower bound in Theorem 4, with a given relevant neighborhood radius $r_U$ and embedding dimension $m$. Note that minimizing its lower bound instead of the exact cost itself is beneficial as it avoids the direct computation of the cost. Recall the lower bound of $W_2(P_U, P_{f^{-1}(V)})$ is (asymptotically) tight (Remark 3) and matches the its upper bound when $n - m \gg 0$. If the lower bound behaves roughly like $W_2(P_U, P_{f^{-1}(V)})$, our simulation result also serves as an empirical evidence that $W_2(P_U, P_{f^{-1}(V)})$ weighs more on high precision.

Specifically, we generate 10000 uniformly distributed samples in a 10-dimensional unit $\ell_2$-ball. We choose $r_U$ such that on average each data point has 500 neighbors inside $B_{r_U}$. We then linearly project these 10 dimensional points into lower dimensional spaces with embedding dimension $m$ from 1 to 9. For each $m$, a different $r_V$ is used to calculate discrete precision and recall. This simulates how optimal $r_V$ according to Wasserstein measure changes with respect to $m$. The result is shown in on the left in Figure 2. Similarly, we can fix $m = 5$ and track optimal $r_V$'s behavior when $r_U$ changes. This is shown on the right in Figure 2.

We evalute our measures based on traditional information retrieval metrics such as f-score. To compute it, we need the discrete/sample-based precision and recall. As discussed in the introduction,

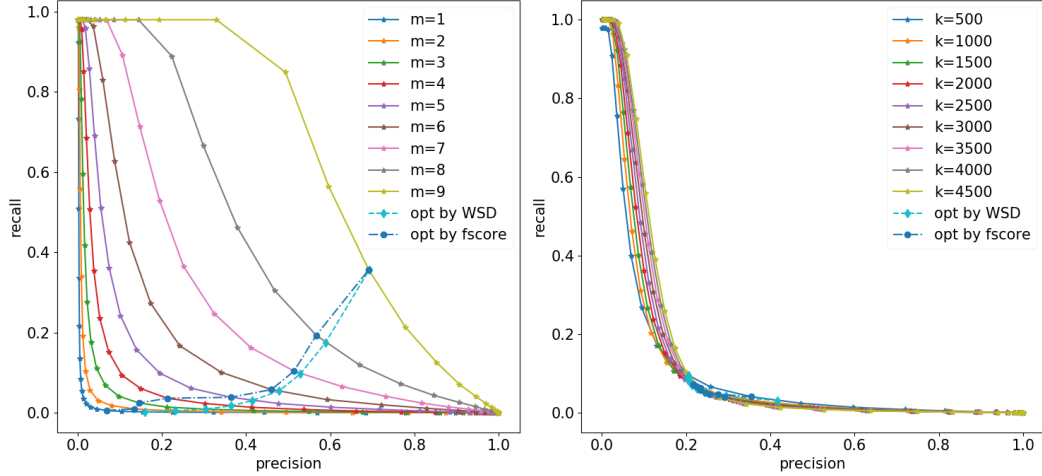

Figure 2: Precision and recall results on uniform samples in a 10 dimensional unit ball. The left figure contains precision-recall curves for a fixed $r_U$ and the optimal $r_V$ is chosen according to $m = 1, \cdots, 9$. The right figure plots the curves for $m = 5$ and the optimal $r_V$'s is chosen for different $r_U$, where $r_U$ is indexed by $k$, the average number of neighbors across all points.

a naive sample based calculations of precision and recall makes $Precision = Recall$ at all times. We compute them alternatively by discretizing Definition 1, by fixing radii $r_U$ and $r_V$. So each $U$ and $f^{-1}(V)$ contain different numbers of neighbors.

$$Precision = \frac{\#(\text{points within } r_U \text{ from } x \text{ and within } r_V \text{ from } y)}{\#(\text{points within } r_V \text{ from } y)} \tag{4}$$

$$Recall = \frac{\#(\text{points within } r_U \text{ from } x \text{ and within } r_V \text{ from } y)}{\#(\text{points within } r_U \text{ from } x)} \tag{5}$$

The optimal $r_V$ according to the lower bound in Theorem 4 (the blue circle-dash-dotted line) aligns closely with the optimal f-score with $\beta = 0.3$ where $\beta$ weighted f-score, also known as f-$\beta$score, is:

$$(1 + \beta^2)\frac{Precision * Recall}{\beta^2 * Precision + recall}.$$

Note that f-score with $\beta < 1$ indeed emphasizes on high precision.

In this provable setting, we have demonstrated our bound's utility. This shows $W_2$ measures' potential for evaluating dimension reduction. In general cases, we won't have such tight lower bounds and it is natural to optimize according to the sample based $W_2$ measures instead. We performed some preliminary experiments on this heuristic, shown in Appendix G.

## 4 Relation to metric space embedding and manifold learning

We lastly situate our work in the lines of research on metric space embedding and manifold learning. One obvious difference between our work and the literature of metric space embedding and manifold learning is that our work mainly focuses on intrinsic dimensionality reduction maps, i.e. $n \gg m$, while in metric space embedding and manifold learning, having $n \leq m < N$ is common.

Our work also differs from the literature of metric space embedding and manifold learning in its learning objective. Learning in these fields aims to preserve the metric structure of the data. Our work attempts to preserve precision and recall, a weaker structure in the sense of embedding dimension (Proposition 2). While they typically look for lowest embedding dimension subject to certain loss (e.g. smoothness, isometry, etc.), in contrast, our learning goal is to minimize the loss (precision and recall etc.) subject to a fixed embedding dimension constraint. In these cases, desired structures will break (Theorem 3) because we cannot choose the embedding dimension $m$ (e.g. for visualizations $m = 2$; for classifications $m$ = number of classes).

We now discuss the technical relations with metric space embedding and manifold learning. Many datasets can be modelled as a finite metric space $\mathcal{M}_k$ with $k$ points. A natural unsupervised learning task is to learn an embedding that approximately preserves pairwise distances. The Bourgain embedding [7] guarantees the metric structure can be preserved with distortion $O(\log k)$ in $l_p^{O(\log^2 k)}$. When the samples are collected in Euclidean spaces, i.e. $\mathcal{M}_k \subset l_2$, the Johnson-Lindenstrauss lemma [10] improves the distortion to $(1 + \epsilon)$ in $l_2^{O(\log(k/\epsilon^2))}$. These embeddings approximately preserve all pairwise distances - global metric structure of $\mathcal{M}_k$ is compatible to the ambient vector space norms. Coming back to our work, it is natural to mimic this approach for precision and recall in $\mathcal{M}_k$. The first problem is that the naive sample based precision and recall are always equal (Section 3.2). A second problem is discrete precision and recall is a non-differentiable objective. In fact, the difficulty of analyzing discrete precision and recall motivates us to look for continuous analogues.

Roughly, our approach is somewhat similar to manifold learning where researchers postulate that the data $\mathcal{M}_k$ are sampled from a continuous manifold $\mathcal{M}$, typically a smooth or Riemannian manifold $\mathcal{M}$ with intrinsic dimension $n$. In this setting, one is interested in embedding $\mathcal{M}$ into $l_2$ locally isometrically. Then one designs learning algorithms that can combine the local information to learn some global structure of $\mathcal{M}$. By relaxing to the continuous cases just like our setting, manifold learning researchers gain access to vast literature in geometry. By the Whitney embedding [25], $\mathcal{M}$ can be smoothly embedded into $\mathbb{R}^{2n}$. By the Nash embedding [35], a compact Riemannian manifold $\mathcal{M}$ can be isometrically embedded into $\mathbb{R}^{p(n)}$, where $p(n)$ is a quadratic polynomial. Hence the task in manifold learning is wellposed: one seeks an embedding $f : \mathcal{M} \subset \mathbb{R}^N \rightarrow \mathbb{R}^m$ with $m \leq 2n \ll N$ in the smooth category or $m \leq p(n) \ll N$ in the Riemannian category. Note that the embedded manifold metrics (e.g. the Riemannian geodesic distances) are not guaranteed to be compatible to the ambient vector space's norm structure with a fixed distortion factor, unlike the Bourgain embedding or the Johnson-Lindenstrauss lemma in the discrete setting. A continuous analogue of the norm compatible discrete metric space embeddings is the Kuratowski embedding, which embeds global-isometrically (preserving pairwise distance) any metric space to an infinite dimensional Banach space $L^\infty$. With $\epsilon$ distortion relaxation, it is possible to embed a compact Riemannian manifold to a finite dimensional normed space. But this appears to be very hard, in that the embedding dimension may grow faster than exponentially in $n$ [30].

Like DR in manifold learning and unlike DR in discrete metric space embedding, rather than global structure we want to preserve local notions such as precision and recall. Unlike DR in manifold learning, since precision and recall are almost equivalent to continuity and injectivity (Theorem 1), we are interested in embeddings in the topological category, instead of the smooth or the Riemannian category. Thus, our work can be considered as manifold learning from the perspective of information retrieval, which leads to the following result.

**Proposition 2.** *If $m \geq 2n$, where $n$ is the dimension of the data manifold $\mathcal{M}$ in domain and $m$ is the dimension of codomain $\mathbb{R}^m$, then there exists a continuous map $f : \mathcal{M} \rightarrow \mathbb{R}^m$ such that $f$ achieves perfect precision and recall for every point $x \in \mathcal{M}$.*

Note that the dimension reduction rate is actually much stronger than the case of Riemannian isometric embedding where the lowest embedding dimension grows polynomially [35]. This is because preserving precision and recall is weaker than isometric embedding. A practical implication is that, we can reduce many more dimensions if we only care about precision and recall.

# 5   Conclusions

We characterized the imperfection of dimensionality reduction mappings from a quantitative topology perspective. We showed that perfect precision and perfect recall cannot be both achieved by any DR map. We then proved a non-trivial upper bound for precision for Lipschitz continuous DR maps. To further quantify the distortion, we proposed a new measure based on $L_2$-Wasserstein distances, and also proved its lower bound for Lipschitz continuous DR maps. It is also interesting to analyse the relation between the recall of a continuous DR map and its modulus of continuity. However, the generality and complexiity of the fibers (inverse images) of these maps so far defy our effort and this problem remains open. Furthermore, it is interesting to develop a corresponding theory in the discrete setting.

**Acknowledgments**

We would like to thank Yanshuai Cao, Christopher Srinivasa, and the broader Borealis AI team for their discussion and support. We also thank Marcus Brubaker, Cathal Smyth, and Matthew E. Taylor for proofreading the manuscript and their suggestions, as well as April Cooper for creating graphics for this work.

## Footnotes

[1] The value of $r_U$ is unknown, and it depends on the user and the input data $x_0$. However, we can assume $r_U$ is small compared to the input domain size. For example, the number of relevant items to a particular user is much fewer than the number of total items.

[2]There is empirical and theoretical evidence that data distribution lies on low dimensional submanifold in the ambient space [27].

[3] Let $A$ be a set. In Euclidean space, $\mathrm{Vol}_n(A) = \mathcal{L}^n(A)$ is the Lebesgue measure. For a general n-rectifiable set, $\mathrm{Vol}_n(A) = \mathcal{H}^n(A)$ is the Hausdorff measure. When $A$ is not rectifiable, $\mathrm{Vol}_n(A) = \mathcal{M}_*^n(A)$ is the lower Minkowski content.

[4]Strictly speaking, the DR maps induced by t-SNE may not be continuous, and hence our theorems do not apply directly. However, since we can measure how closely parametric t-SNE (which is continuous) behaves as t-SNE and there is empirical evidence to their similarity [21], our theorems may apply again.

[5]Such attempts existed in literature on visualization of dimensionality reduction (e.g. [34]). However, since these works are based on heuristics, it is less clear what they measure, nor do they enjoy theoretical guarantee.

[6]An antecedent of this problem was studied in Section 2.3 of [24], where the authors optimize over the more restricted class of ellipses with fixed area. For our purpose, the minimization is over bounded measurable sets.

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
