[Supplementary Material]

# A  Waist Inequality and Isoperimetric Inequality

**Theorem 7** (Waist Inequality, Akopyan and Karasev [1]). *Let $m \leq n$ and $f$ be a continuous map from the ball $B_R^n$ of radius $R$ to $\mathbb{R}^m$. Then there exists some $y \in \mathbb{R}^m$ such that*

$$Vol_{n-m}\left(f^{-1}(y)\right) \geq Vol_{n-m}\left(B_R^{n-m}\right).^7$$

*Moreover, for all $\epsilon > 0$:*

$$Vol_n\left(f^{-1}(y) \oplus \epsilon\right) \geq \frac{1}{2\pi R} Vol_{n-m+1}\left(S_R^{n-m+1}\right) Vol_m\left(B_1^m\right) p^m(\epsilon), \qquad (6)$$

*where $p^m(\epsilon)$ is $\epsilon^m\left(1 + o(1)\right)$, i.e. $\lim_{\epsilon \to 0} \frac{p^m(\epsilon)}{\epsilon^m} = 1$, and $f^{-1}(y) \oplus \epsilon$ denotes the set of points $x \in B_R^n$ such that $d(x, f^{-1}(y)) < \epsilon$, $S_R^{n-m+1}$ is the (n+m-1)-dimensional sphere of radius R, and $B_1^m$ is the unit $m$ ball.*

**Remark 4.** *When $m = 1$, Waist Inequality generalizes classic concentration of measure on $B_R^n$, which says most volume of a high dimensional ball concentrates around its equator slab, as $n \to \infty$. When $m > 1$, we can roughly interpret the theorem as $f^{-1}(y) \oplus \epsilon$ is big in $n - m$ dimensions in the sense of volume, thus it generalizes concentration of measure when $m > 1$.*

Intuitively the Waist inequality states that a higher dimensional space is too big in the sense of **volume** that we cannot hope to squeeze it **continuously** into lower dimensional spaces, without collapsing in some direction(s). In other words, if an input domain is higher dimensional and thus in some sense large, then it must be large in at least one direction. Waist inequality is a precise quantitative version of the topological invariance of dimension, which states balls of different dimensions cannot be homeomorphically mapped to each other. It is this mis-match between high and low dimensional nature of volumes that motivates us to formulate and prove the imperfection between precision and recall. A recent survey of the inequality can be found in [15].

**Theorem 8** (Isoperimetric Inequality). *Suppose $U \subset \mathbb{R}^n$ is a bounded (Hausdorff) measurable set, with (Hausdorff) $n - 1$ measurable boundary, denoted as $Vol_{n-1}\partial U$. Then:*

$$Vol_n(U) = Vol_n(B_1^n) \implies Vol_{n-1}(\partial U) \geq Vol_{n-1}(\partial B_1^n)$$

*Stated differently,*

$$Vol_n(U) \leq \frac{1}{n^{\frac{n}{n-1}} Vol_n(B_1)^{\frac{1}{n-1}}} Vol_{n-1}(\partial U)^{\frac{n}{n-1}}$$

The first way of looking at the isoperimetric inequality is from an optimization viewpoint. It states that Euclidean balls are optimal sets in terms of minimizing the $n - 1$ hypersurface volume, with a constraint on their $n$ volume. The second (equivalent) inequality is from an inequality angle. It allows us to control the $n$ volume of a set in terms of its boundary's $n - 1$ volume. For more information about this fundamental inequality, we refer the reader to [28].

Among all equal volume sets on the plane, the isoperimetric inequality says that the disc has the least perimeter. This statement compares all domains to balls. The waist inequality is its close cousin with perhaps stronger topological flavor. This is a statement about all continuous maps $f : B_R^n \to \mathbb{R}^m$: we can find $f^{-1}(y)$ such that $Vol_{n-m}(f^{-1}(y)) \geq Vol_{n-m}B_R^{n-m}$. This compares all continuous maps's volume-maximal fiber to balls. See Fig. 3 for an illustration in 3D.

# B  Precision, Recall, One-To-One, and Continuity

We extend the definitions of continuity and injectivity to allow exceptions on a measure zero set. For a dimensionality reduction map $f : \mathbb{R}^n \to \mathbb{R}^m$, we say it is essentially one-to-one if its 'injectivity' is essentially no more than the reduction part. The manifold setting $f : \mathcal{M}^n \to \mathbb{R}^m$ is handled naturally by using coordinates and parametrization by open sets in $\mathbb{R}^n$, as in classical differential topology and differential geometry.

DISK                                        SURFACE

Figure 3: The above pictorial illustration compares $f^{-1}(y)$ - the pancake surface living in a 3-ball to a disc living in the 3-ball. We see that $f^{-1}(y)$ has bigger or equal area than the corresponding disc.

**Definition 2** (Essential Continuity). *$f$ is essentially continuous at $x$, if for any $\epsilon > 0$, there exists $r > 0$, such that for all the neighbourhood $U \ni x$ satisfying $diam(U) \leq r$,*

$$\text{Vol}_n\left(\{u \in U \ : \ |f(u) - f(x)| > \epsilon\}\right) = 0.$$

*We say $f$ is essentially continuous on a set $W$ if $f$ is essentially continuous at every $w \in W$.*

**Definition 3** (Essential Injectivity). *$f$ is essentially one-to-one or essentially injective at $x$, if for $y = f(x) \in \mathbb{R}^m$, $\text{Vol}_{n-m}\left(f^{-1}(y)\right) = 0$[8]. $f$ is essentially one-to-one on a set $W$ if $f$ is essentially one-to-one at every $w \in W$.*

Note that the definition of essential continuity (one-to-one, respectively) strictly generalizes the definition of continuity (one-to-one, respectively). In other words, every continuous function is essentially continuous, and there exists discontinuous functions that are essentially continuous. The following lemma shows that if $f$ is essentially continuous on an open set $W$, then $f$ is continuous on $W$.

**Lemma 1** (Essential continuity in a neighborhood). *Essential continuity in a neighborhood and continuity in a neighborhood are equivalent.*

*Proof.* It is sufficient to prove that if $f$ is essentially continuous on an open set $W$, then $f$ is continuous on $W$. Assume that $f$ is not continuous on $W$, i.e., there exists $\eta > 0$, $w \in W$ and a sequence $\{w_1, \dots, w_n, \dots\}$ such that $\lim_{n \to \infty} w_n = w$, but $|f(w_n) - f(w)| \geq \eta$. Since $f$ is essentially continuous on $W$, there exists a neighbourhood of $w$, $U \subset W$, such that $\text{Vol}_n(E_U) = 0$, where $E_U = \{u \in U \ : \ |f(u) - f(w)| > \eta/3\}$. Note that for large enough $M$, $w_M \in E_U$. Moreover, since $f$ is also essentially continuous at $w_M$, for a small neighbourhood $V$ of $w_M$, $\text{Vol}_n(\{v \in V \ : \ |f(v) - f(w_M)| \leq \eta/3\}) = \text{Vol}_n(V) > 0$. However, note that this positive measure set $\{v \in V \ : \ |f(v) - f(w_M)| \leq \eta/3\}$ is a subset of $E_U$ by the definition of $E_U$, contradicting $\text{Vol}_n(E_U) = 0$. $\square$

We next prove the equivalence between perfect recall and essential continuity.

**Proposition 3.** *For any map $f : \mathcal{M} \subset \mathbb{R}^N \to \mathbb{R}^m$, $f$ achieves perfect recall in an open set $W$, if and only if $f$ is essentially continuous on $W$.*

*Proof.* **(Perfect Recall $\Rightarrow$ Essential Continuity)** For any $x \in W$, any $\epsilon > 0$, let $V = \{f(v) \in \mathbb{R}^m \ : \ |f(v) - f(x)| \leq \epsilon\}$. Since $f$ achieves perfect recall at $x$, there exists $r > 0$, such that $\text{Vol}_n(f^{-1}(V) \cap B_r(x)) = \text{Vol}_n(B_r(x))$. Therefore, for any $U$ such that $U \subset B_r(x)$,

$$\text{Vol}_n\left(\{u \in U \ : \ |f(u) - f(x)| > \epsilon\}\right) \leq \text{Vol}_n\left(\{u \in U \ : \ u \notin f^{-1}(V) \cap B_r(x)\}\right) = 0.$$

Thus $f$ is essentially continuous at $x$.

**(Essential Continuity $\Rightarrow$ Perfect Recall)** By Lemma 1, $f$ is continuous on $W$. For any $x \in W$, assume $f(x) = y$. For any $r_V > 0$, $f^{-1}(B_{r_V}(y))$ is an open set in $\mathcal{M}$. Therefore, there exists small enough $r_U$ such that $B_{r_U}(x) \subset f^{-1}(V)$, thus $\text{Recall}^f(B_{r_U}(x), B_{r_V}(y)) = 1$. $\square$

Based on this proposition, we can further prove that if $f$ is (essentially) continuous on $W$, then $f$ has neither perfect precision nor essential injectivity property on $W$.

**Proposition 4.** *Let $f : \mathcal{M}^n \subset \mathbb{R}^N \to \mathbb{R}^m$, with $m < n$. If $f$ is (essentially) continuous with approximate differential well defined on an open set $W$ almost everywhere, [9], then $f$ possesses neither perfect precision nor essential injectivity on $W$.*

*Proof.* **(Continuous in neighborhood $\Rightarrow$ Not Essentially Injective)** We first prove that if $f$ is continuous on $W \subset \mathbb{R}^n$, then $f$ is not essentially one-to-one on $W$. To prove that $f$ does not have perfect precision, it is sufficient to prove that the perfect precision of $f$ implies $f$ being essentially one-to-one. We handle the manifold case at the end of the proof, by coordination: $\phi : U \subset \mathcal{M}^n \to V \subset \mathbb{R}^n$, and parametrization $\phi^{-1} : V \subset \mathbb{R}^n \to U \subset \mathcal{M}^n$.

Assume $f$ is essentially one-to-one on $W$, thus for any $y \in f(W) \subset \mathbb{R}^m$,

$$\text{Vol}_{n-m}(f^{-1}(y)) = \int_{f^{-1}(y)} d\text{Vol}_{n-m}(p) = 0.$$

Since $W \subset \mathbb{R}^n$ is open, there is an open ball $B_\tau^n \subset W$ such that we can consider the restriction of $f$ onto $B_\tau^n$. Now Theorem 7 guarantees the existence of $y_\tau \in f(B_\tau^n)$ such that

$$\text{Vol}_{n-m}(f^{-1}(y_\tau)) \geq \text{Vol}_{n-m}(B_\tau^n) > 0.$$

This contradiction completes the proof in the Euclidean case.

Now, for a map $f : W \subset \mathcal{M}^n \to \mathbb{R}^m$. We consider the restriction of $f$ on $U \subset W$ where $U$ is homeomorphic to $\mathbb{R}^n$. Then the composite map: $f \circ \phi^{-1} \to \mathbb{R}^m$ is again a map between Euclidean spaces. The argument above applies and we complete this part of the proof.

**(Perfect Precision $\Rightarrow$ Essential One-to-one)** Assume that $f$ is not essentially one-to-one on $W$, thus $f$ is not one-to-one on $W$. Therefore, there exist $y$, $z_1$, and $z_2$ such that $f(z_1) = f(z_2) = y$. Without loss of generality, assume $d(z_1, z_2) = 1$. Since $f$ has perfect precision, picking $U = B_{0.4}^m(z_1)$, there exists $r_{V,1}$, such that $\text{Vol}_n\left(f^{-1}(B_r^m(y)) \cap B_{0.4}^m(z_1)\right) = \text{Vol}_n\left(f^{-1}(B_r^m(y))\right)$ for $r \leq r_{V,1}$. Similarly, there exists $r_{V,2}$, such that $\text{Vol}_n\left(f^{-1}(B_r^m(y)) \cap B_{0.4}^m(z_2)\right) = \text{Vol}_n\left(f^{-1}(B_r^m(y))\right)$ for $r \leq r_{V,2}$. Further note that $B_{0.4}^m(z_1) \cap B_{0.4}^m(z_2) = \emptyset$. For $r \leq \min\{r_{V,1}, r_{V,2}\}$, then

$$\text{Vol}_n(f^{-1}(B_r^m(y))) \geq \text{Vol}_n(f^{-1}(B_r^m(y)) \cap B_{0.4}^m(z_1)) + \text{Vol}_n(f^{-1}(B_r^m(y)) \cap B_{0.4}^m(z_2))$$
$$= 2 * \text{Vol}_n(f^{-1}(B_r^m(y))).$$

Therefore, $\text{Vol}_n\left(f^{-1}(B_r^m(y))\right) = 0$. Now since $f$ is continuous, $f^{-1}(B_r^m(y))$ is an open set in $\mathcal{M}$, thus $\text{Vol}_n\left(f^{-1}(B_r^m(y))\right)$ cannot be 0, a contradiction. $\square$

Based on Propositions 3 and 4, the proof of Theorem 1 is straightforward.

*Proof of Theorem 1.* It is sufficient to prove that if $f$ achieves perfection recall at $W$, then $f$ cannot achieve perfect precision at $W$. Since $f$ achieves perfect recall at $W$, by Proposition 3 $f$ is continuous, thus by Proposition 4 $f$ cannot achieve perfect precision at $W$. $\square$

# C  Proof of Theorem 2

We present the proof of Theorem 2 in this section. The following proposition develops a lower bound for the volume of the inverse image of $f$ on a particular small open set.

**Proposition 5.** *If $f$ is a continuous function with Lipschitz constant $L$, then for any $y \in \mathbb{R}^m$ and $\epsilon > 0$,*

$$Vol_n \left( f^{-1}(B_\epsilon^m(y)) \right) \geq Vol_n \left( f^{-1}(y) \oplus \frac{\epsilon}{L} \right).$$

*Proof.* Since $f$ is Lipschitz, for any $x$ such that $\mathrm{d}(x, f^{-1}(y)) \leq \frac{\epsilon}{L}$, $|f(x) - f(y)| \leq \epsilon$. Thus

$$f^{-1}(y) \oplus \frac{\epsilon}{L} = \{x \in \mathcal{M} : \mathrm{d}(x, f^{-1}(y)) \leq \frac{\epsilon}{L}\} \subset \{x \in \mathcal{M} : |f(x) - f(y)| \leq \frac{\epsilon}{L}\} = f^{-1}\left(B_\epsilon^m(y)\right).$$

Therefore,

$$\mathrm{Vol}_n \left( f^{-1}(y \oplus \epsilon) \right) \geq \mathrm{Vol}_n \left( f^{-1}(y) \oplus \frac{\epsilon}{L} \right).$$

$\square$

*Proof of Theorem 2.* By Theorem 7, there exists $y \in \mathbb{R}^m$ such that

$$\mathrm{Vol}_n \left( f^{-1}(y) \oplus \epsilon \right) \geq \frac{1}{2\pi R} \mathrm{Vol}_{n-m+1} \left( S_R^{n-m+1} \right) \mathrm{Vol}_m \left( B_1^m \right) \epsilon^m \left( 1 + o(1) \right).$$

For any $x \in f^{-1}(y)$, $r_U, r_V > 0$, recall that $\mathrm{Precision}^f(U, V) = \frac{\mathrm{Vol}_n(f^{-1}(V) \cap U)}{\mathrm{Vol}_n(f^{-1}(V))} \leq \frac{\mathrm{Vol}_n(U)}{\mathrm{Vol}_n(f^{-1}(V))}$, thus a lower bound of $\mathrm{Vol}_n(f^{-1}(V))$ leads to an upper bound for $\mathrm{Precision}^f(U, V)$. Further note that

$$
\begin{aligned}
\mathrm{Vol}_n(f^{-1}(V)) &= \mathrm{Vol}_n \left( f^{-1}(y \oplus r_V) \right) \\
&\geq \mathrm{Vol}_n \left( f^{-1}(y) \oplus (r_V/L) \right) \\
&\geq \frac{1}{2\pi R} \mathrm{Vol}_{n-m+1}(S^{n-m+1}) \mathrm{Vol}_m(B_1^m) R^{n-m+1} p^m(r_V/L) \\
&= \frac{\pi^{(n-m)/2}}{\Gamma(\frac{n-m}{2} + 1)} \frac{\pi^{m/2}}{\Gamma(\frac{m}{2} + 1)} R^{n-m} p^m(r_V/L),
\end{aligned}
\tag{7}
$$

where the first inequality is due to Proposition 5, the second inequality is due to the Waist Inequality Equation (6), and $p^m(x) = x^m (1 + o(1))$. Combining the volume calculation on $U$,

$$
\begin{aligned}
\mathrm{Precision}^f(U, V) &\leq \frac{\frac{\pi^{n/2}}{\Gamma(\frac{n}{2}+1)} r_U^n}{\frac{\pi^{n-m/2}}{\Gamma(\frac{n-m}{2}+1)} \frac{\pi^{m/2}}{\Gamma(\frac{m}{2}+1)} R^{n-m} p^m(r_V/L)} \\
&\leq \frac{\Gamma(\frac{n-m}{2} + 1) \Gamma(\frac{m}{2} + 1)}{\Gamma(\frac{n}{2} + 1)} (\frac{r_U}{R})^{n-m} \frac{r_U^m}{p^m(r_V/L)}.
\end{aligned}
$$

$\square$

Theorem 2 generalizes as long as there is a corresponding waist theorem for that space. And roughly the condition of having a waist theorem is that a space is 'truly' $n$ dimensional. We therefore conjecture that Theorem 2 holds in various settings in machine learning where we are dealing with truly $n$ dimensional data. In the rest of this section, we are going to prove analogues of Theorem 2 under the non-Euclidean norm.

We define the necessary concepts first. In the non-Eucldiean case, the generalized unit ball is a convex body.

**Definition 4** (Generalized Unit Ball, e.g. Wang [36])**.** *Let $p_1, p_2, \ldots, p_n \geq 1$. A generalized unit $n$ ball is defined as the following convex body:*

$$B_{p_1, p_2, \ldots, p_n} = \{(x_1, x_2, \ldots, x_n) : |x_1|^{p_1} + \ldots + |x_n|^{p_n} \leq 1\} \tag{8}$$

**Theorem 9** (Volume of Generalized Ball, Wang [36])**.**

$$Vol_n B_{p_1, p_2, \ldots, p_n} = 2^n \frac{\Gamma(1 + 1/p_1) \ldots \Gamma(1 + 1/p_n)}{\Gamma(1 + 1/p_1 + \ldots + 1/p_n)} \tag{9}$$

**Definition 5** (Log-Concave Measure). *A Borel measure $\mu$ on $\mathbb{R}^n$ is log-concave if for any compacts sets $A \subset \mathbb{R}^n$ and $B \subset \mathbb{R}^n$, and for any $0 < \lambda < 1$:*

$$\mu(\lambda A \oplus (1 - \lambda)B) \geq \mu(A)^\lambda \mu(B)^{1-\lambda} \tag{10}$$

**Theorem 10** (Brunn-Minkowski Inequality). *Let $Vol_n$ denote Lebesgue measure on $\mathbb{R}^n$. Let $A$ and $B$ be two nonempty compact subsets of $\mathbb{R}^n$. Then:*

$$[Vol_n(A \oplus B)]^{1/n} \geq [Vol_n(A)]^{1/n} + [Vol_n(B)]^{1/n} \tag{11}$$

The following lemma is well known in concentration of measure and convex geometry. We prove it here for completeness.

**Lemma 2** (Lebesgue Measure on Convex Sets is Log-Concave). *Let $Vol_n$ denote Lebesgue measure on $\mathbb{R}^n$. The (induced) restricted measure, $Vol_n$, by restricting $Vol_n$ to any convex sets is log-concave.*

*Proof.* Plugging $\lambda A$ and $(1 - \lambda)B$ to theorem 10, we have:

$$\text{Vol}_n^{1/n}(\lambda A \oplus (1 - \lambda)B) \geq \text{Vol}_n^{1/n}(\lambda A) + \text{Vol}_n^{1/n}((1 - \lambda)B) \tag{12}$$

$$= \lambda \text{Vol}_n^{1/n}(A) + (1 - \lambda)\text{Vol}_n^{1/n}(B) \tag{13}$$

$$\geq \text{Vol}_n^{\lambda/n}(A)\text{Vol}_n^{(1-\lambda)/n}(B) \tag{14}$$

where the first equality follows because the $\lambda$ (or $1 - \lambda$ respectively) is scaled be a factor or $\lambda^n$ and taking $n$th root gives the equality, and the last inequality follows from the weighted arithmetic-geometric mean inequality. Raising to the $n$th power, we get:

$$\text{Vol}_n(\lambda A \oplus (1 - \lambda)B) \geq \text{Vol}_n^\lambda(A)\text{Vol}_n^{(1-\lambda)}(B) \tag{15}$$

To finish the proof, we note that for any $A$ and $B$ as nonempty compact subsets of a convex set $K \subset \mathbb{R}^n$ in the Euclidean space, the Lebesgue measures restricted on $K$, $\text{Vol}_n(A)$ and $\text{Vol}_n(B)$ can be written as Lebegues measures on $A$ and $B$. Convexity of $K$ ensures $\lambda A \oplus (1 - \lambda)B$ is still in the set $K$. $\qquad\square$

To deduce an analogue of Theorem 2, we need the following waist inequality for log-concave measures.

**Theorem 11** (Waists of Arbitrary Norms, Theorem 5.4 of Akopyan and Karasev [2]). *Suppose $K \subset \mathbb{R}^n$ is a convex body, $\mu$ a finite log-concave measure supported on $K$, and $f : K \longrightarrow \mathbb{R}^m$ is continuous. Then for any $\epsilon \in [0, 1]$ there exists $y \in \mathbb{R}^m$ such that:*

$$\mu(f^{-1}(y) \oplus \epsilon K) \geq \epsilon^m \mu(K) \tag{16}$$

**Proposition 6** (Precision on Arbitrarilly Normed Balls). *Let $m < n$. Let $f : B_{R;p_1,p_2,\ldots,p_n} \longrightarrow \mathbb{R}^m$ be a $L$-Lipschtiz continuous map defined on a generalized $n$ ball with radius $R$ from Definition 4. Let $r_U$ and $r_V$ be radii of two generalized balls, with dimensions $n$ and $m$ respectively. Then there exists $y$ depending on $r_V$ such that:*

$$Prec^f(U, V) \leq (\frac{r_U}{R})^{n-m}(\frac{r_U}{r_V/L})^m \tag{17}$$

*Proof.* We would like to apply theorem 11. Since $B_{R;p_1,p_2,\ldots,p_n}$ is a convex body, Lebesgue meaure $\text{Vol}_n$ on $B_{R;p_1,p_2,\ldots,p_n}$ is log-concave by Lemma 2. Then by Theorem 11, for $r_V/L$, there exists $y \in \mathbb{R}^m$ such that:

$$\text{Vol}_n(f^{-1}(y) \oplus \frac{r_V}{L}K) \geq (\frac{r_V}{L})^m\text{Vol}_n(K) \tag{18}$$

where $K = B_{R;p_1,p_2,\ldots,p_n}$. Now by Proposition 5,

$$\text{Vol}_n(f^{-1}(V)) = \text{Vol}_n(f^{-1}(B_{r_V;p_1,p_2,\ldots,p_n})) \geq (\frac{r_V}{L})^m\text{Vol}_n(K) \tag{19}$$

Therefore:

$$Prec^f(U,V) \leq \frac{\text{Vol}_n(U)}{\text{Vol}_n(f^{-1}(V))} \tag{20}$$

$$\leq \frac{\text{Vol}_n(B_{r_U;p_1,p_2,\ldots,p_n})}{\text{Vol}_n(B_{R;p_1,p_2,\ldots,p_n})(\frac{r_V}{L})^m} \tag{21}$$

$$= \frac{2^n \frac{\Gamma(1+1/p_1)\ldots\Gamma(1+1/p_n)}{\Gamma(1+1/p_1+\ldots+1/p_n)} r_U^{n-m} r_U^m}{2^n \frac{\Gamma(1+1/p_1)\ldots\Gamma(1+1/p_n)}{\Gamma(1+1/p_1+\ldots+1/p_n)} R^{n-m}(\frac{r_V}{L})^m} \tag{22}$$

$$= (\frac{r_U}{R})^{n-m}(\frac{r_U}{r_V/L})^m \tag{23}$$

$$\square$$

## D    Proof of Theorem 3

The proof of Theorem 3 is based on the idea that the fibers of certain type of continuous DR maps are mostly 'large'. A map $f$ has a large fiber at $y$ if $f^{-1}(y)$'s volume is lower bounded by that of a linear map. This concept of 'large' fiber is actually an essential concept in the proof of the waist inequality. The intuition we try to capture is that fibers of $f$ are considered big if their $n-m$ volumes are comparable to that of a surjective linear map.

The next two theorems show that for either of the following cases:

- $m = 1$; or
- $f : B_R^n \to \mathbb{R}^m$ be a $k$-layer neural network map with Lipschitz constant $L$, whose linear layers are surjective.

the fibers of $f$ are mostly 'large'.

**Theorem 12** (Average Waist Inequality for Balls, m = 1)**.** *Let $f$ be a continuous map from $B_R^n$ to $\mathbb{R}$, and $\tau = Vol_{n+1}\left(Proj^{-1}(y) \oplus \epsilon\right)$ for an arbitrary $y \in Proj(S_R^{n+1})$, then for all $\epsilon > 0$*

$$Vol_n\left(\left\{z \in B_R^n \ : \ Vol_n\left(f^{-1}(f(z)) \oplus \epsilon\right) \geq \frac{1}{2\pi R}\tau\right\}\right)$$

$$\geq \frac{1}{2\pi R}Vol_{n+1}\left(\left\{x \in S_R^{n+1} : Vol_{n+1}\left(Proj^{-1}(Proj(x)) \oplus \epsilon\right) \geq \tau\right\}\right).$$

**Proposition 7.** *Let $f$ be a $k$ layer neural network with nonlinear activations (ReLu, LeakyReLu, tanh, etc.) from $B_R^n$ to $(0,1)^m$ and Proj be an arbitrary linear projection on $B_R^n$. Then for any $\tau$ the following inequality holds,*

$$Vol_n\left(\left\{x \in B_R^n \ : \ Vol_n\left(f^{-1}(f(x)) \oplus \epsilon\right) \geq \tau\right\}\right)$$

$$\geq Vol_n\left(\left\{x \in B_R^n : Vol_n\left(Proj^{-1}(Proj(x)) \oplus \epsilon\right) \geq \tau\right\}\right).$$

The proof of Theorem 12 is postponed to Appendix D.1, while the proof of Proposition 7 is postponed to Appendix D.2. We are now ready to derive a bound on DR maps' **average-case performance** over the domain based on Theorem 12 and Proposition 7.

*Proof of Theorem 3.* We only present the proof when $f : B_R^n \to \mathbb{R}^m$ is a $k$-layer neural network map with Lipschitz constant $L$ by Proposition 7. The other case can be proved similarly by Theorem 12.

Given any $y \in \text{Proj}(B_R^n)$, pick $\tau = \text{Vol}_n\left(\text{Proj}^{-1}(y) \oplus \epsilon\right)$. By Proposition 7 for all $\epsilon > 0$,

$$\text{Vol}_n\left(\left\{x \in B_R^n \ : \ \text{Vol}_n\left(f^{-1}(f(x)) \oplus \epsilon\right) \geq \text{Vol}_n\left(\text{Proj}^{-1}(y) \oplus \epsilon\right)\right\}\right) \tag{24}$$

$$\geq \text{Vol}_n\left(\left\{x \in B_R^n : \text{Vol}_n\left(\text{Proj}^{-1}(\text{Proj}(x)) \oplus \epsilon\right) \geq \text{Vol}_n\left(\text{Proj}^{-1}(y) \oplus \epsilon\right)\right\}\right).$$

Since Proj is a linear map, we have

$$\text{Vol}_n\left(\left\{x \in B_R^n : \text{Vol}_n\left(\text{Proj}^{-1}(\text{Proj}(x)) \oplus \epsilon\right) \geq \text{Vol}_n\left(\text{Proj}^{-1}(y) \oplus \epsilon\right)\right\}\right)$$

$$= \text{Vol}_n\left(\left\{x \in B_R^n : \text{Vol}_{n-m}\left(\text{Proj}^{-1}(\text{Proj}(x))\right) \geq \text{Vol}_{n-m}\left(\text{Proj}^{-1}(y)\right)\right\}\right).$$

Further note that $\mathrm{Proj}^{-1}(y)$ is an $n-m$ ball with radius $r(y) = \sqrt{R^2 - \|y\|^2}$. Thus,

$$\mathrm{Vol}_n\left(\left\{x \in B_R^n : \mathrm{Vol}_{n-m}\left(\mathrm{Proj}^{-1}\left(\mathrm{Proj}(x)\right)\right) \geq \mathrm{Vol}_{n-m}\left(\mathrm{Proj}^{-1}(y)\right)\right\}\right)$$

$$= \int_{B_{\|y\|}^m} \mathrm{Vol}_{n-m}(\mathrm{Proj}^{-1}(t))\mathrm{d}t.$$

Therefore,

$$\mathrm{Vol}_n\left(\left\{x \in B_R^n : \mathrm{Vol}_n\left(f^{-1}\left(f(x)\right) \oplus \epsilon\right) \geq \mathrm{Vol}_n\left(\mathrm{Proj}^{-1}(y) \oplus \epsilon\right)\right\}\right) \geq \int_{B_{\|y\|}^m} \mathrm{Vol}_{n-m}(\mathrm{Proj}^{-1}(t))\mathrm{d}t. \tag{25}$$

Lastly, pick $y$ such that $\|y\| = \sqrt{R^2 - r_U^2 - \delta^2}$, so $\mathrm{Proj}^{-1}(y)$ has radius $\sqrt{r_U^2 + \delta^2}$. Let $\mathcal{E}$ denote the event $\mathrm{Vol}_n\left(f^{-1}\left(f(x)\right) \oplus \epsilon\right) \geq \mathrm{Vol}_n\left(B_{\sqrt{r_U^2+\delta^2}}^{n-m} \oplus \epsilon\right)$, thus

$$\mathbb{P}(\mathcal{E}) \geq \frac{\int_{B_{\sqrt{R^2-u^2-\delta^2}}^m} \mathrm{Vol}_{n-m}\mathrm{Proj}^{-1}(t)\mathrm{d}t}{\mathrm{Vol}_n(B_R^n)}.$$

The remaining proof is almost identical to the proof of Theorem 2. Under the event $\mathcal{E}$,

$$\mathrm{Vol}_n(f^{-1}(V)) = \mathrm{Vol}_n\left(f^{-1}(y \oplus r_V)\right)$$

$$\geq \mathrm{Vol}_n\left(f^{-1}(y) \oplus (r_V/L)\right)$$

$$\geq \mathrm{Vol}_{n-m}(B_1^{n-m})\mathrm{Vol}_m(B_1^m)(\sqrt{r_U^2+\delta^2})^{n-m}(r_V/L)^m$$

$$= \frac{\pi^{(n-m)/2}}{\Gamma(\frac{n-m}{2}+1)}\frac{\pi^{m/2}}{\Gamma(\frac{m}{2}+1)}\sqrt{r_U^2+\delta^2}^{n-m}(r_V/L)^m, \tag{26}$$

where the first inequality is due to Proposition 5, the second inequality is due to the event $\mathcal{E}$. Combining the volume calculation on $U$,

$$\mathrm{Precision}^f(U,V) \leq \frac{\frac{\pi^{n/2}}{\Gamma(\frac{n}{2}+1)}r_U^n}{\frac{\pi^{n-m/2}}{\Gamma(\frac{n-m}{2}+1)}\frac{\pi^{m/2}}{\Gamma(\frac{m}{2}+1)}\sqrt{r_U^2+\delta^2}^{n-m}(r_V/L)^m}$$

$$\leq \frac{\Gamma(\frac{n-m}{2}+1)\Gamma(\frac{m}{2}+1)}{\Gamma(\frac{n}{2}+1)}(\frac{r_U}{\sqrt{r_U^2+\delta^2}})^{n-m}\frac{r_U^m}{(r_V/L)^m}.$$

$\square$

## D.1 Proof of Theorem 12

The proof uses the following average waist inequality for spheres. Let $P : S_R^{n+1} \longrightarrow B_R^n$ be the orthogonal projection, $\sigma_R$ and $\nu_R$ denote the corresponding Hausdorff measures on $S_R^{n+1}$ and $B_R^n$. Further, let $\mathrm{Proj} : S_R^{n+1} \to \mathbb{R}$ be the restriction to $S_R^{n+1}$ of a surjective linear map $\widehat{\mathrm{Proj}} : \mathbb{R}^{n+2} \to \mathbb{R}$.

**Theorem 13** (Average Waist Inequality for Spheres [3]). *Let $f$ be a continuous map from $S_R^{n+1}$ to $\mathbb{R}$, then for all $y \in Proj(S_R^{n+1})$, we have:*

$$Vol_{n+1}\{x \in S_R^{n+1} : Vol_{n+1}(f^{-1}(f(x)) \oplus \epsilon) \geq Vol_{n+1}(Proj^{-1}(y) \oplus \epsilon)\}$$

$$\geq$$

$$Vol_{n+1}\{x \in S_R^{n+1} : Vol_{n+1}(Proj^{-1}(Proj(x)) \oplus \epsilon) \geq Vol_{n+1}(Proj^{-1}(y) \oplus \epsilon)\},$$

*where*

$$Vol_{n+1}\left(Proj^{-1}(y) \oplus \epsilon\right) = 2\pi Vol_n\left(S_{R_{Proj^{-1}(y)}}^n\right) Vol_1\left(B_1^1\right)\left(p^1(\epsilon)\right),$$

*$p^1(\epsilon)$ is $\epsilon(1 + o(1))$, i.e. $\lim_{\epsilon \to 0}\frac{p^1(\epsilon)}{\epsilon} = 1$, and $f^{-1}(y) \oplus \epsilon$ denotes the set of points $x \in S_R^{n+1}$ such that $d(x, f^{-1}(y)) < \epsilon$, $S_R^n$ is the n-dimensional sphere of radius $R$, and $S_{R_{Proj^{-1}(y)}}^n$ is the sphere with radius $R_{Proj^{-1}(y)}$ depending on where $y$ is taken in $f(S_R^{n+1})$, i.e. $R_{Proj^{-1}(y)}^2 = R^2 - y^2$.*

We are going to adapt the proof technique of theorem 1 from [1], by replacing the existential waist inequality (7) with its average version - theorem 13. We need the following lemma:

**Lemma 3** ( Orthogonal Projection e.g. Akopyan and Karasev [1] ). *Let $P : S_R^{n+1} \longrightarrow B_R^n$ be the orthogonal projection. Then $P$ is $1$ - Lipschitz and $P_\# \sigma_R = 2\pi R \nu_R$. In other words, $P$ sends the uniform Hausdorff measure $\sigma$ in $S_R^{n+1}$ to the uniform Lebesgue measure $\nu_n$ in $B_R^n$ up to constant $2\pi R$.*

*Proof of Theorem 12.* Given a map $f : B_R^n \longrightarrow \mathbb{R}$, consider $\hat{f} = f \circ P : S_R^{n+1} \to \mathbb{R}$, where $P$ is the orthogonal projection. By Lemma 3, $P$ is 1-Lipschitz, thus for any $y \in \mathbb{R}$,

$$P^{-1} \left( f^{-1}(y) \right) \oplus \epsilon \subset P^{-1} \left( f^{-1}(y) \oplus \epsilon \right) \Rightarrow \text{Vol}_{n+1} \left( \hat{f}^{-1}(y) \oplus \epsilon \right) \leq \text{Vol}_{n+1} \left( P^{-1} \left( f^{-1}(y) \oplus \epsilon \right) \right). \tag{27}$$

Further, since $P_\# \sigma_R = 2\pi R \nu_R$,

$$\text{Vol}_{n+1} \left( P^{-1} \left( f^{-1}(y) \oplus \epsilon \right) \right) = 2\pi R \text{Vol}_n \left( f^{-1}(y) \oplus \epsilon \right). \tag{28}$$

Combining Equations (27) and (28), for $\tau \in \mathbb{R}$,

$$\left\{ x \in S_R^{n+1} : \hat{f}(x) = y, \text{Vol}_{n+1} \left( \hat{f}^{-1}(y) \oplus \epsilon \right) \geq \tau \right\}$$
$$\subset \left\{ x \in S_R^{n+1} : \hat{f}(x) = y, \text{Vol}_n \left( f^{-1}(y) \oplus \epsilon \right) \geq \frac{\tau}{2\pi R} \right\}. \tag{29}$$

Similarly, by $P_\# \sigma_R = 2\pi R \nu_R$,

$$\text{Vol}_{n+1} \left( \left\{ x \in S_R^{n+1} : \hat{f}(x) = y, \text{Vol}_n \left( f^{-1}(y) \oplus \epsilon \right) \geq \frac{\tau}{2\pi R} \right\} \right)$$
$$= 2\pi R \text{Vol}_n \left( \left\{ z \in B_R^n : f(z) = y, \text{Vol}_n \left( f^{-1}(y) \oplus \epsilon \right) \geq \frac{\tau}{2\pi R} \right\} \right). \tag{30}$$

Thus by combining Equations (29) and (30), we have

$$\text{Vol}_{n+1} \left( \left\{ x \in S_R^{n+1} : \hat{f}(x) = y, \text{Vol}_{n+1} \left( \hat{f}^{-1}(y) \oplus \epsilon \right) \geq \tau \right\} \right)$$
$$\leq 2\pi R \text{Vol}_n \left( \left\{ z \in B_R^n : f(z) = y, \text{Vol}_n \left( f^{-1}(y) \oplus \epsilon \right) \geq \frac{\tau}{2\pi R} \right\} \right)$$

Finally, note that $\hat{f}$ meets the condition in theorem 13. Thus for all $y \in \text{Proj}(S_R^{n+1})$:

$$\text{Vol}_{n+1} \left( \left\{ x \in S_R^{n+1} : \text{Vol}_{n+1} \left( \text{Proj}^{-1} \left( \text{Proj}(x) \right) \oplus \epsilon \right) \geq \text{Vol}_{n+1} \left( \text{Proj}^{-1}(y) \oplus \epsilon \right) \right\} \right)$$
$$\leq \text{Vol}_{n+1} \left( \left\{ x \in S_R^{n+1} : \text{Vol}_{n+1} \left( \hat{f}^{-1} \left( \hat{f}(x) \right) \oplus \epsilon \right) \geq \text{Vol}_{n+1} \left( \text{Proj}^{-1}(y) \oplus \epsilon \right) \right\} \right)$$
$$\leq 2\pi R \text{Vol}_n \left( \left\{ z \in B_R^n : \text{Vol}_n \left( f^{-1} \left( f(z) \right) \oplus \epsilon \right) \geq \frac{1}{2\pi R} \text{Vol}_{n+1} \left( \text{Proj}^{-1}(y) \oplus \epsilon \right) \right\} \right).$$

$\square$

## D.2    Proof of Proposition 7

We first prove that Proposition 7 holds for any surjective linear map.

**Proposition 8.** *Let $f$ be any surjective linear map (PCA, linear neural networks) from $B_R^n$ to $\mathbb{R}^m$, and Proj be an arbitrary surjective linear projection from $B_R^n$ to $\mathbb{R}^m$. Then for any $\tau$ the following inequality holds,*

$$Vol_n \left( \left\{ x \in B_R^n : Vol_n \left( f^{-1} \left( f(x) \right) \oplus \epsilon \right) \geq \tau \right\} \right)$$
$$\geq Vol_n \left( \left\{ x \in B_R^n : Vol_n \left( Proj^{-1} \left( Proj(x) \right) \oplus \epsilon \right) \geq \tau \right\} \right).$$

*Proof.* By the singular value decomposition, any linear dimension reduction map $f$ can be decomposed as a composition or unitary operators ($U_m$ and $V_n$), signed dialation of full rank ($\Sigma$), and

projection operator of rank $m$ $(\widehat{\text{Proj}})$, where $\widehat{\text{Proj}}$ linearly projects from $\mathbb{R}^n$ to $\mathbb{R}^m$ (or more commonly $\Sigma \circ \widehat{\text{Proj}}$ is called rectangular diagonal matrix map): $f = U_m \circ \Sigma \circ \widehat{\text{Proj}} \circ V_n^*$. The set

$$\left\{ x \in B_R^n \ : \ \text{Vol}_n \left( f^{-1} \left( f(x) \right) \oplus \epsilon \right) \geq \tau \right\}$$

$$= \left\{ x \in B_R^n \ : \ \text{Vol}_n \left( (U_m \circ \Sigma \circ \widehat{\text{Proj}} \circ V_n^*)^{-1} \left( U_m \circ \Sigma \circ \widehat{\text{Proj}} \circ V_n^*(x) \right) \oplus \epsilon \right) \geq \tau \right\}$$

$$= \left\{ x \in B_R^n \ : \ \text{Vol}_n \left( (V_n^*)^{-1} \circ \widehat{\text{Proj}}^{-1} \circ \Sigma^{-1} \circ U_m^{-1} \circ U_m \circ \Sigma \left( \widehat{\text{Proj}} \circ V_n^*(x) \right) \oplus \epsilon \right) \geq \tau \right\}$$

$$= \left\{ x \in B_R^n \ : \ \text{Vol}_n \left( (V_n^*)^{-1} \circ \widehat{\text{Proj}}^{-1} \circ \left( \widehat{\text{Proj}} \circ V_n^*(x) \right) \oplus \epsilon \right) \geq \tau \right\}$$

$$= \left\{ x \in B_R^n \ : \ \text{Vol}_n \left( V_n \circ \widehat{\text{Proj}}^{-1} \circ \left( \widehat{\text{Proj}} \circ V_n^*(x) \right) \oplus \epsilon \right) \geq \tau \right\}$$

$$= \left\{ x \in B_R^n \ : \ \text{Vol}_n \left( V_n \circ \widehat{\text{Proj}}^{-1} \circ \left( \widehat{\text{Proj}}(x) \right) \oplus \epsilon \right) \geq \tau \right\}$$

$$= \left\{ x \in B_R^n \ : \ \text{Vol}_n \left( \widehat{\text{Proj}}^{-1} \circ \widehat{\text{Proj}}(x) \oplus \epsilon \right) \geq \tau \right\},$$

where the last two equalities follow because unitary operator $V_n^*$ and $V_n$ don't affect volumes because they are linear isometries. We note this shows the distribution of fiber volume is the same for any surjective linear map. Finally, note that by symmetry,

$$\left\{ x \in B_R^n \ : \ \text{Vol}_n \left( \widehat{\text{Proj}}^{-1} \circ \widehat{\text{Proj}}(x) \oplus \epsilon \right) \geq \tau \right\} = \left\{ x \in B_R^n \ : \ \text{Vol}_n \left( \text{Proj}^{-1} \circ \text{Proj}(x) \oplus \epsilon \right) \geq \tau \right\}.$$

$\square$

**Lemma 4** (Monotonicity of Fiber Volume under Compositions). *Let $f : B_R^n \longrightarrow X$ and $g : X \longrightarrow \mathbb{R}^m$ be any maps for some set $X$. Then for any $\tau$ we have the following inequality:*

$$Vol_n \left( \left\{ x \in B_R^n \ : \ Vol_n \left( (f \circ g)^{-1} \left( f \circ g(x) \right) \oplus \epsilon \right) \geq \tau \right\} \right)$$

$$\geq Vol_n \left( \left\{ x \in B_R^n : Vol_n \left( f^{-1} \left( f(x) \right) \oplus \epsilon \right) \geq \tau \right\} \right).$$

*Proof.* Consider: $a \in \left\{ x \in B_R^n : \text{Vol}_n \left( f^{-1} \left( f(x) \right) \oplus \epsilon \right) \geq \tau \right\}$ and we let $b = f(a)$. We obviously have $b \in g^{-1} \circ g(b)$. Therefore $a \in f^{-1}(b) \subset f^{-1} \circ g^{-1} \circ g(f(a))$. Thus,

$$\left\{ x \in B_R^n : \text{Vol}_n \left( f^{-1} \left( f(x) \right) \oplus \epsilon \right) \geq \tau \right\} \subset \quad \left\{ x \in B_R^n \ : \ \text{Vol}_n \left( (f \circ g)^{-1} \left( f \circ g(x) \right) \oplus \epsilon \right) \geq \tau \right\}.$$

$\square$

*Proof of Proposition 7.* We proceed by induction on $k$. When $k = 1$, it is given by lemma 4, by noting a one layer net is a composition of any activation with a surjective linear map, $L_1$. Assume this is true for a $k$ layer neural net, $f_k$, with $k$ layers such that $k \geq 1$. So we have:

$$\text{Vol}_n \left( \left\{ x \in B_R^n \ : \ \text{Vol}_n \left( f_k^{-1} \left( f_k(x) \right) \oplus \epsilon \right) \geq \tau \right\} \right)$$

$$\geq \text{Vol}_n \left( \left\{ x \in B_R^n : \text{Vol}_n \left( \text{Proj}^{-1} \left( \text{Proj}(x) \right) \oplus \epsilon \right) \geq \tau \right\} \right).$$

We need to check a neural net $f_{k+1}$ with $k + 1$ layers: $f_{k+1} = \tanh \circ L_{k+1} \circ f_k$. But this is again a composition between functions and we can apply Lemma 4. This completes the proof. $\square$

In light of Proposition 8, we can characterize $\text{Proj}_1^{-1}(t)$ and $\text{Proj}_2^{-1}(t)$ explicitly. Since the bound holds for any surjective linear map, we can choose in particular $\text{Proj}_1^{-1}(t)$ and $\text{Proj}_2^{-1}(t)$ to be the coordinate projection from $\mathbb{R}^n$ to $\mathbb{R}^m$ (with all eigenvalues equal to 1). Then $t = (t_1, \cdots, t_m) \in B_R^m$, $\text{Proj}_1^{-1}(t) = S_{\Re_1}^{n-m+1}$ and $\text{Proj}_2^{-1}(t) = B_{\Re_2}^{n-m}$, where $\Re_1 = \Re_2 = \sqrt{R^2 - \sum_{i=1}^m t_i^2}$.

# E  Proofs for Section 3

This section is devoted to the proofs for Section 3. We first present the proof of Theorem 4.

*Proof of Theorem 4.* By Equation (7),

$$\text{Vol}_n(f^{-1}(V)) \geq \frac{\pi^{n/2}}{\Gamma(\frac{n-m}{2}+1)\Gamma(\frac{m}{2}+1)} R^{n-m} p^m (r_V/C).$$

Let $B_{r^\#}$ be the ball with the same volume as $\text{Vol}_n(f^{-1}(V))$ and a common center with $U$. Thus

$$r^\# \geq r = \left(\frac{\Gamma(\frac{n}{2}+1)}{\Gamma(\frac{n-m}{2}+1)\Gamma(\frac{m}{2}+1)}\right)^{\frac{1}{n}} R^{\frac{n-m}{n}} (p^m (r_V/C))^{\frac{1}{n}}. \tag{31}$$

By Theorem 5,

$$W_2^2(\mathbb{P}_U, \mathbb{P}_{f^{-1}(V)}) \geq W_2^2(\mathbb{P}_U, \mathbb{P}_{B_{r^\#}}) = \int_{B_r(u)} |x - T(x)|^2 \, d\mathbb{P}_{B_{r^\#}}(x),$$

thus it is sufficient to lower bound the last term. Under the condition that $\text{Vol}_n(f^{-1}(V)) \geq \text{Vol}_n(U)$,

$$\int_{B_{r^\#}} |x - T(x)|^2 \, d\mathbb{P}_{B_{r^\#}}(x) = \int_{B_{r^\#}} |x - \frac{r_U}{r^\#} x|^2 \, d\mathbb{P}_{B_{r^\#}}(x)$$
$$= \left(1 - \frac{r_U}{r^\#}\right)^2 \int_{B_{r^\#}} |x|^2 \, d\mathbb{P}_{B_{r^\#}}(x).$$

Further,

$$\int_{B_{r^\#}} |x|^2 \, d\mathbb{P}_{B_{r^\#}}(x) = \int_0^{r^\#} r^2 \frac{1}{\text{Vol}_n(f^{-1}(V))} dS^{n-1}(r) dr$$
$$= \frac{1}{\text{Vol}_n(f^{-1}(V))} \frac{2\pi^{n/2}}{\Gamma(\frac{n}{2})} \int_0^{r^\#} r^{n+1} dr$$
$$= \frac{n}{n+2} (r^\#)^2.$$

Therefore,

$$W_2^2(\mathbb{P}_U, \mathbb{P}_{f^{-1}(V)}) \geq \left(1 - \frac{r_U}{r^\#}\right)^2 \frac{n}{n+2} (r^\#)^2 = \frac{n}{n+2} (r^\# - r_U)^2.$$

Note that the above lower bound is monotonically increasing with respect to $r^\#$ for $r^\# > r_U$. Therefore from Equation (31), when $r > r_U$, replacing $r^\#$ by $r$ gives a lower bound for $W_2^2(\mathbb{P}_U, \mathbb{P}_{f^{-1}(V)})$.

Further, note that as $n \to \infty$, $r \to R$, we have:

$$W_2^2(\mathbb{P}_U, \mathbb{P}_{f^{-1}(V)}) = \Omega\left((R - r_U)^2\right).$$

$\square$

The rest of this section is to prove Theorem 5. The key step is to show the following lemma.

**Lemma 5** (Reduction to Optimal Partial Transport). *Given $f(x) = 1/\mathcal{V} \leq 1/Vol(B_r)$, the optimal distribution $f_M$ for the optimal transport problem*

$$\min_{\mathbb{P}\,:\,\mathbb{P}\text{ is dominated by } f} W_2(\mathbb{P}, \mathbb{P}_{B_r}) \tag{32}$$

*is the uniform distribution over $B_{r^\#}$ where $r^\#$ is the radius such that $Vol(B_{r^\#}) = \mathcal{V}$.*

By Lemma 5, let $f(x) = 1/\mathcal{V}$, the optimal solution for the problem

$$\inf_{W\,:\,\text{Vol}_n(W)=\mathcal{V}} W_2(\mathbb{P}_U, \mathbb{P}_W) = W_2(\mathbb{P}_U, \mathbb{P}_{B_r})$$

is the same as support of the optimizer of Equation (32), thus proving the first statement of Theorem 5.

The proof of Lemma 5 is based on the uniqueness of the optimal transport map for the optimal partial transport problem [9, 11]. We summarize the statements in [11][10] as a theorem here for completeness.

**Theorem 14** (Figalli [11]). *Let $f, g \in L^1(B_R^n)$ be two nonnegative functions, and denote by $\Xi_{\leq}(f, g)$ the set of nonnegative finite Borel measures on $B_R^n \times B_R^n$ whose first and second marginals are dominated by $f$ and $g$ respectively, i.e. $\xi(A \times B_R^n) \leq \int_A f(x)dx$ and $\xi(B_R^n \times A) \leq \int_A g(y)dy$, for all Borel $A \subset B_R^n$. Denote $\mathscr{M}(\xi) := \int_{B_R^n \times B_R^n} d\xi$ and fix $M \in [\|\min(f(x), g(x))\|_{L_1}, \min(\|f\|_{L_1}, \|g\|_{L_1})]$. Then there exists a unique optimizer $\xi_M$[11] to the following optimal partial transport problem:*

$$\inf_{\xi \in \Xi_{\leq}(f,g); \mathscr{M}(\xi)=M} C(\xi) = \inf_{\xi \in \Xi_{\leq}(f,g); \mathscr{M}(\xi)=M} \int_{B_R^n \times B_R^n} |x-y|^2 d\xi(x,y)$$

*Moreover, there exist Borel sets $A_1, A_2 \subset B_R^n$ such that $\xi_M$ has left and right marginals whose densities $f_M = 1_{A_1} f$ and $g_M = 1_{A_2} g$ are given by the restrictions of $f$ and $g$ to $A_1$ and $A_2$ respectively, where $1_A$ denotes characteristic function on the set $A$.*

*Finally, there exists a unique optimal transport map $T$[12], such that*

$$\min_{\xi \in \Xi_{\leq}(f,g); \mathscr{M}(\xi)=M} C(\xi) = \int_{B_R^n} |T(x) - x|^2 df_M(x),$$

*where $f_M$ is the marginal of $\xi_M$ over the first $B_R^n$.*

We will prove Lemma 5 in two different ways. The first is based on calculus and reducing the problem to one dimensional optimal transport. The second one utilizes the extreme points property that characterizes the densities $f_M = 1_{A_1} f$ and $g_M = 1_{A_2} g$ (Proposition 3.2 and Theorem 3.3 in [18]). [13]

*Proof of Lemma 5, first approach.* Let $\mathcal{V} = \text{Vol}(W)$, define $f(x) = 1/\mathcal{V}$ be a constant function on $B_R^n$ and $g(x) = \frac{1}{\text{Vol}(B_r)}$ if $x \in B_r$ and 0 otherwise. Also, let $M = 1$. solving the problem

$$\min_{\mathbb{P} : \mathbb{P} \text{ is dominated by } f} W_2(\mathbb{P}, \mathbb{P}_{B_r}) \tag{33}$$

is equivalent to solving the following optimal partial transport problem

$$\inf_{\xi \in \Xi_{\leq}(f,g); \mathscr{M}(\xi)=1} C(\xi) = \inf_{\xi \in \Xi_{\leq}(f,g); \mathscr{M}(\xi)=1} \int_{B_R^n \times B_R^n} |x-y|^2 d\xi(x,y). \tag{34}$$

In particular, since $\text{Vol}(B_R^n) \geq \mathcal{V} > \text{Vol}(B_r)$, it is straightforward to see that $\|\min(f(x), g(x))\|_{L_1} = \text{Vol}(B_r)/\mathcal{V} < 1$, and $\min(\|f(x)\|_{L_1}, \|g(x)\|_{L_1}) \geq 1$. By Theorem 14, the optimization problem $\inf_{\xi \in \Xi_{\leq}(f,g); \mathscr{M}(\xi)=1} C(\xi)$ has a unique solution $\xi^*$. Now given $\xi^*$, the optimal solution $\mathbb{P}^*$ of Equation (33) and $\mathbb{P}_{B_r}$ are the first and the second marginals of $\xi^*$. Thus it is sufficient to prove that the first marginal of $\xi^*$ is a uniform distribution.

Let $f_M$ be the first marginal of $\xi^*$ and $g_M = g$ be the second marginal. We first show that $f_M$ is rotationally invariant. To see that, for any rotation map $\mathcal{R}$, note that $\mathcal{R}(B_R^n) = B_R^n$, $\mathcal{R}(B_r) = B_r$, $f \circ \mathcal{R} = f$, and $g \circ \mathcal{R} = g$. Therefore, $f_M \circ \mathcal{R}$ is the unique optimal solution for the optimization problem

$$\inf_{\xi \in \Xi_{\leq}(f \circ \mathcal{R}, g \circ \mathcal{R}); \mathscr{M}(\xi)=1} \int_{\mathcal{R}(B_R^n) \times \mathcal{R}(B_R^n)} |x-y|^2 d\xi(x,y) = \inf_{\xi \in \Xi_{\leq}(f,g); \mathscr{M}(\xi)=1} \int_{B_R^n \times B_R^n} |x-y|^2 d\xi(x,y).$$

Thus, $f_M \circ \mathcal{R} = f_M$, i.e. $f_M$ is rotationally invariant, up to a measure zero set. For a density function to be rotationally invariant, it is straightforward that its support $S$ is also rotationally invariant, thus is a union of $(n-1)$ spheres. Similarly, one can also prove that $T$ is equivariant under rotations.

We next prove that $f_M$ is a uniform distribution. Note that $g_M$ is a uniform distribution over $B_r$. Define $\hat{G}(t)$ to be the the cumulative distribution $\hat{g}$ for $g_M$ in the polar coordinate marginalized on the sphere, i.e.,

$$\hat{G}(t) = \int_0^t \frac{1}{\text{Vol}_n B_r^n} \text{Vol}_{n-1}(S_u^{n-1}) du,$$

for every $0 \le t \le r$, and $G(t) = 1$ for $t > r$. Similarly, since $f_M$ is also rotationally invariant, we can also define its cumulative distribution in the polar coordinate marginalized on the sphere. Note that $\mathrm{d}\mu_{f_M} = f_M(x)\mathrm{d}S_r^{n-1}\mathrm{d}r$, let $\hat{f}(r) = \int f_M(x)\mathrm{d}S_r^{n-1}$, thus

$$F(B_t) = \int_{B_t} f_M(x)\mathrm{d}S_u^{n-1}\mathrm{d}u = \int_0^t \int f_M(x)\mathrm{d}S_u^{n-1}\mathrm{d}u = \int_0^t \hat{f}(u)\mathrm{d}u = \hat{F}(t).$$

Finally, note that $T$ is also rotationally invariant, thus $W_2(f_M, \mathbb{P}_{B_r}) = W_2(\hat{f}, \hat{g})$. It is sufficient to prove that $\hat{f}(u) = \mathrm{Vol}_{n-1}(S_u^{n-1})/\mathcal{V}$, thus by rotationally invariant $f_M(x) = 1/\mathcal{V}$ is a uniform distribution.

Note that $\hat{F}(t) \le \hat{G}(t)$ and $\hat{f}(u) = \int f_M(x)\mathrm{d}S_u^{n-1} \le \int f(x)\mathrm{d}S_u^{n-1} = \mathrm{Vol}_{n-1}(S_u^{n-1})/\mathcal{V}$. By a reformulation of the one dimensional Wasserstein distance [45]:

$$W_2(\hat{f}, \hat{g}) = \int_0^1 |\hat{F}^{-1}(t) - \hat{G}^{-1}(t)|^2 \mathrm{d}t$$

$$= \int_0^{r^{\#}} |x - \hat{G}^{-1}\left(\hat{F}(x)\right)|^2 \mathrm{d}\hat{F}(x), \tag{35}$$

which is just the area between between the graphs of $\hat{F}(r)$ and $\hat{G}(r)$. It is straightforward that the optimal $\hat{f}$ will maximize the growth rate of $\hat{F}$ in order to minimize the area, i.e. $\hat{f}(u) = \int f(x)\mathrm{d}S_u^{n-1} = 1/\mathcal{V}\mathrm{Vol}_{n-1}(S_u^{n-1})$. Therefore, $f_M(x) = 1/\mathcal{V}$ is a uniform distribution over $B_{r^{\#}}$ where $r^{\#}$ is the radius of $B_{r^{\#}}$ such that $\mathrm{Vol}(B_{r^{\#}}) = \mathcal{V}$. $\qquad\square$

*Proof of Lemma 5, second approach.* The proof starts in exactly the same way as in the first approach, up to the rotational invariance part. Instead of using the polar coordinate argument, we directly apply by invoking the second statement in Theorem 14, so $f_M = 1_{A_1} f$. But we know that $f(x) = 1/\mathcal{V}$ is a uniform distribution, and the claim follows. $\qquad\square$

Further note that by Equation (35), the optimal transport from $\hat{F}$ to $\hat{G}$ is

$$\hat{T}(u) = \hat{G}^{-1}\left(\hat{F}(u)\right) = \hat{G}^{-1}\left(\frac{1}{\mathcal{V}}\mathrm{Vol}_n(B_u^n)\right) = \left(\frac{\mathrm{Vol}_n(B_{r_U})}{\mathcal{V}}\right)^{1/n} u = \frac{r_U}{r_{\mathcal{V}}}r,$$

for $0 \le r \le r_M$. Note that $T$ is rotationally symmetric, thus the optimal transport $T(x) = \frac{r_U}{r_{\mathcal{V}}}x$, for $x \in B_{r_{\mathcal{V}}}$

Lastly, it remains to prove

$$\inf_{W: \mathrm{Vol}_n(W) \ge \mathcal{V}} W_2(\mathbb{P}_U, \mathbb{P}_W) = \inf_{W: \mathrm{Vol}_n(W) = \mathcal{V}} W_2(\mathbb{P}_U, \mathbb{P}_W),$$

which follows the next lemma.

**Lemma 6** (Monotonicity of Volume Comparison)**.** *Given two balls $B_{r_1}$ and $B_{r_2}$ such that $Vol(B_{r_1}) \ge Vol(B_{r_2})$, then for any $A \subset \mathbb{R}^n$ such that $Vol(A) \ge Vol(B_{r_1})$,*

$$W_2(\mathbb{P}(A), \mathbb{P}(B_{r_2})) \ge W_2(\mathbb{P}(B_{r_1}), \mathbb{P}(B_{r_2})).$$

*Proof of Lemma 6.* We have shown that $W_2(\mathbb{P}(A), \mathbb{P}(B_{r_2})) \ge W_2(\mathbb{P}(B_{r_A}), \mathbb{P}(B_{r_2}))$, where $B_{r_A}$ is a ball with Volume $\mathrm{Vol}(A)$. It remains to prove that

$$W_2(\mathbb{P}(B_{r_A}), \mathbb{P}(B_{r_2})) \ge W_2(\mathbb{P}(B_{r_1}), \mathbb{P}(B_{r_2}))$$

Let $T_A(x) = \frac{r_2}{r_A} x$, and $T_1(x) = \frac{r_2}{r_1} x$. By Theorem 14,

$$
\begin{aligned}
W_2^2(\mathbb{P}(B_{r_A}^n), \mathbb{P}(B_{r_2}^n)) &= \int_{B_R^n} |x - T_A(x)|^2 \mathrm{d}\mathbb{P}_{B_{r_A}^n} \\
&= \int_{B_R^n} \left| x - \frac{r_2}{r_A} x \right|^2 \mathrm{d}\mathbb{P}_{B_{r_A}^n} \\
&= \int_{B_R^n} \left( 1 - \frac{r_2}{r_A} \right)^2 |x|^2 \, \mathrm{d}\mathbb{P}_{B_{r_A}^n} \\
&\geq \int_{B_R^n} \left( 1 - \frac{r_2}{r_1} \right)^2 |x|^2 \, \mathrm{d}\mathbb{P}_{B_{r_A}^n} \\
&= \int_{B_R^n} |x - T_1(x)|^2 \mathrm{d}\mathbb{P}_{B_{r_A}^n} \\
&= W_2^2(\mathbb{P}(B_{r_1}^n), \mathbb{P}(B_{r_2}^n))
\end{aligned}
$$

$\square$

To make Theorem 5 complete, it remains to investigate the remaining cases when $0 < \mathcal{V} < \mathrm{Vol}_n(U)$.

*Proof.* We claim that when $0 < \mathcal{V} < \mathrm{Vol}(U)$, $\inf_{W \colon \mathrm{Vol}_n(W) = \mathcal{V}} W_2(\mathbb{P}_U, \mathbb{P}_W) = 0$, and it is not attained by any set. Let $\mathrm{Vol}_n(W_k) = \mathcal{V}$ and keep $W_k \subset U$ such that the mass of $W_k$ is evenly distributed among the intersection between successively finer rectangular grids and $U$. Inside each intersection, the two distributions have the same probability mass. Since both are uniform probability distributions, their densities scale inversely proportional to their support sizes inside the intersection. Each little intersection is inside a little cube with width $\frac{2R}{k}$. We take $\xi$ to be the product measure between $\mathbb{P}(U)$ and $\mathbb{P}(W)$. Now, when we compute:

$$
W_2(\mathbb{P}_U, \mathbb{P}_W) = \inf_{\xi \in \Xi(\mathbb{P}_U, \mathbb{P}_W)} \mathbb{E}_{(a,b) \sim \xi} [\|a - b\|_2^2]^{1/2} \leq \mathbb{E}_{(a,b) \sim \xi} [\|a - b\|_2^2]^{1/2}.
$$

The integrand $\|a - b\|_2^2 \leq \sqrt{n} \frac{2R}{k}$. By letting $k \to \infty$ (finer grids), we see that $\inf_{W \colon \mathrm{Vol}_n(W) = \mathcal{V}} W_2(\mathbb{P}_U, \mathbb{P}_W) = 0$.

However, the infimum is not attained by any set $W$ with $\mathrm{Vol}_n(W) = \mathcal{V} < \mathrm{Vol}_n(U)$. Without loss of generality, we assume $W \subset U$. Then $\mathrm{Vol}_n(U - W) > 0$. So $W_2(\mathbb{P}_U, \mathbb{P}_W) > 0$.

$\square$

# F    Proofs for Section 4

We prove the proposition 2 here. We begin with a lemma.

**Lemma 7** (One-To-One $\implies$ Perfect Precision). *Let $\mathcal{M}$ be a Riemannian manifold. Let $f : \mathcal{M} \to \mathbb{R}^m$ be an open map. Then $f$ achieves perfect precision.*

*Proof.* $f$ is an open map, mapping open sets to open sets. For every $U \subset \mathcal{M}$, $f(U)$ is open in $\mathbb{R}^m$. Since $f(U)$ is open and contains $y = f(x)$, there exists $r_V > 0$ such that $V \subset f(U)$. This implies $f^{-1}(V) \subset U$. But then $\mathrm{Precision}^f(U, V) = \frac{\mathrm{Vol}_n(f^{-1}(V) \cap U)}{\mathrm{Vol}_n(f^{-1}(V))} = 1$ for such $V$ and $U$. $\square$

*Proof of Proposition 2.* Let $\mathcal{M}$ be an $n$-dimensional Riemannian manifold and $m \geq 2n$ be the embedding dimension. By the Whitney embedding theorem, there exists a smooth map $f$ such that $f(\mathcal{M})$ embeds into $\mathbb{R}^m$. Thus $f$ is an open map from $\mathcal{M}$ to $f(\mathcal{M})$. We now apply lemma 7 to arrive at the conclusion. $\square$

## G  Wasserstein many-to-one, discontinuity and cost

In general, we do not have theoretical lower bound for $W_2$ measure. It is natural to use the sample based Wasserstein distances as substitutes. We perform some preliminary study of this heuristics below.

Recall Wasserstein distance is the minimal cost for mass-preserving transportation between regions. The Wasserstein $L^2$ distance is:

$$W_2(\mathbb{P}_a, \mathbb{P}_b) = \inf_{\xi \in \Xi(\mathbb{P}_a, \mathbb{P}_b)} \mathbb{E}_{(a,b) \sim \xi}[\|a - b\|_2^2]^{1/2} \tag{36}$$

where $\Xi(\mathbb{P}_a, \mathbb{P}_b)$ denotes all joint distributions $\xi(a, b)$ whose marginal distributions are $\mathbb{P}_a$ and $\mathbb{P}_b$. Intuitively, among all possible ways of transporting the two distributions, it looks for the most efficient one. With the same intuition, we use Wasserstein distance between $U$ and $f^{-1}(V)$[14] to measure precision (See Section 3.2). This not only captures similar overlapping information as the setwise precision: $\frac{Vol_n(f^{-1}(V) \cap U)}{Vol_n(f^{-1}(V))}$, but also captures the shape differences and distances between $U$ and $f^{-1}(V)$. Similarly, Wasserstein distance between $f(U)$ and $V$ may capture the degree of discontinuity. $W_2(\mathbb{P}_{f(U)}, \mathbb{P}_V)$ **captures continuity** and $W_2(\mathbb{P}_U, \mathbb{P}_{f^{-1}(V)})$ **captures injectivity**.

In practice, we calculate Wasserstein distances between two groups of samples, $\{a_i\}$ and $\{b_j\}$, using algorithms from [6][15]. Specifically, we solve

$$\min_m \sum_i \sum_j d_{i,j} m_{i \to j} \; ,$$
$$\text{such that}: \; m_{i \to j} \geq 0, \; \sum_i m_{i \to j} = 1, \; \sum_j m_{i \to j} = 1, \tag{37}$$

where $d_{i,j}$ is the distance between $a_i$ and $b_j$ and $m_{i \to j}$ is the mass moved from $a_i$ to $b_j$. When $\{a_i\} \subset U$ and $\{b_j\} \subset f^{-1}(V)$, it is **Wasserstein many-to-one**. When $\{a_i\} \subset f(U)$ and $\{b_j\} \subset V$, it is **Wasserstein discontinuity**. High many-to-one likely implies low precision, and high discontinuity likely implies low recall. The average of many-to-one and discontinuity is **Wasserstein cost**. An implementation of these measures can be found at `github.com/BorealisAI/eval_dr_by_wsd`.

We note that our measures bypass some practical difficulties on using precision and recall as evaluation measures. The first issue was discussed in Section 3.2, where we discussed that precision and recall are always equal when computed naively. This defeats their very purpose for capture both continuity and injectivity. Computing them based on Equation (4) and Equation (5) is more sensible, but it introduces another difficulty in practice due to high dimensionality: the radii $r_U$ and/or $r_V$ need to be quite large in order for some (outlier data point) $x$ to have a reasonable number of neighboring data points. Some $x$ ends up having many neighboring points, while others have very few[16]. This introduces a high variance on the number of neighboring data points across $x$. Our Wasserstein measures bypass both practical issues: having a fixed number of neighbors won't make $W_2(\mathbb{P}_{f(U)}, \mathbb{P}_V)$ and $W_2(\mathbb{P}_U, \mathbb{P}_{f^{-1}(V)})$ equal. In our experiments, we choose 30 neighboring points for all of $U$, $f^{-1}(V)$, $f(U)$ and $V$.

### G.1  Preliminary experiments on Wasserstein Measures, Compare Visualization Maps

In this section, we show preliminary results on using Wasserstein measures directly (instead of its lower bound) to choose between dimensionality reduction algorithms. We may interpret this as choosing between different information retrieval systems in the DR visualization context. Figure 4 and 5 show the visualization results of 5 different methods on the S-curve and Swiss roll toy datasets respectively. These include PCA, multidimensional scaling (MDS) [39], locally linear embedding (LLE) [43], Isomap [33] and t-SNE [22]. In the results of PCA and MDS, the mappings squeeze the original data into narrower regions in the 2D projection space. Squeezing naturally implies high degree of many-to-one. At the same time, PCA mapping is linear, the MDS mapping in this case is close to linear, which makes both PCA and MDS has a low discontinuity. For S-curve and

Figure 4: quality of different methods on S-curve

Swiss roll, LLE, Isomap and t-SNE all works well in the sense that they successfully unwrapped the manifold. However, when local compression or stretch happens, the Wasserstein discontinuity and many-to-one will will increase slightly. For example, in the S-curve LLE results, the right side of data is compressed. Therefore it has a slightly higher many-to-one value, while the discontinuity is still low.

Figure 6 shows the visualization results on MNIST digits. As a linear map, PCA still has a relatively lower discontinuity and higher degree of many-to-one. MDS preserve global distances, at the cost of sacrificing local distances. thus can map nearby points to far away locations, at the same time mapping far a way points together has poor local one-to-one property. So it has both high discontinuity and many-to-one on MNIST digits. Compared with the previous toy example, LLE and Isomap both have a significant performance drop. Among all the methods, t-SNE still have the best local properties for MNIST digits, due to its neighborhood preservation objective.

Figure 5: quality of different methods on Swiss roll

## G.2 Preliminary experiments precision and recall (continuity v.s. injectivity) tradeoff

Theorem 1 suggests there is a trade-off between precision and recall, or equivalently continuity v.s. injectivity, via Proposition 1. In this section, we attempt to illustrate this tradeoff phenomenon by altering the degree of continuity of a DR algorithm in a practical situation. We choose t-SNE on MNIST because: 1) Heuristically t-SNE's perplexity parameter controls the degree of continuity: a higher perplexity means more neighboring data points will contract together and contraction is a continuous map (respectively, lower perplexity creates more tearing and spliting); 2) the tradeoff may be best seen through DR algorithms that operate at the optimal tradeoff level. t-SNE has proved itself as the de facto standard for visualization in various datasets; 3) As a practical dataset, MNIST visualization is still simple enough that humans can inspect and diagnose.

Fig. 7 shows visualizations with different t-SNE perplexity parameter. Each row is indexed by a different perplexity (perp $= 2, 8, \cdots, 1024$), with the intuition that the t-SNE DR map becomes more continuous with larger perplexity. The middle two columns are colored by our Wasserstein

Figure 6: quality of different methods on MNIST

measures, with lower discontinuity costs representing more continuous maps (higher recall) and lower many-to-one costs indicating more injective (higher precision) maps. The precision and recall tradeoff can be observed in the perplexity ranging from 32 to 128. As t-SNE becomes more continuous, it is also less injective. In this range, inspection by eye suggests t-SNE gives good visualizations.

Outside of the range of $(32, 128)$ both precision and recall become worse. We interpret this as t-SNE is giving relatively bad visualizations for these choices of parameter, as can be inspected by eye. For example, when perplexity $= 512$ and $1024$, t-SNE actually tends to have lower recall while precision worsens. When perplexity $< 32$, it is less clear whether it is due to: 1) there is a tradeoff but our measures do not capture it. Our neighborhood size is also 30 (comparable or bigger than the perplexity), so the scale may not be fair (on the other hand, choosing neighborhood size smaller than 30 may introduce very high variance in the estimation); 2) t-SNE actually performances worse on both continuity and injectivity, reflected by our measures. By inspection on the visualization, we believe it is probably because t-SNE isn't performing at any optimal level, so tradeoff cannot be seen.

Figure 7: quality of t-SNE with different perplexities on MNIST

## Supplementary References

[1] Arseniy Akopyan and Roman Karasev. A tight estimate for the waist of the ball. *Bulletin of the London Mathematical Society*, 49(4):690–693, 2017.

[3] Hannah Alpert and Larry Guth. A family of maps with many small fibers. *Journal of Topology and Analysis*, 7(01):73–79, 2015.

[39] Ingwer Borg and Patrick JF Groenen. *Modern multidimensional scaling: Theory and applications*. Springer Science & Business Media, 2005.

[9] Luis A Caffarelli and Robert J McCann. Free boundaries in optimal transport and Monge-Ampére obstacle problems. *Annals of mathematics*, 171:673–730, 2010.

[11] Alessio Figalli. The optimal partial transport problem. *Archive for rational mechanics and analysis*, 195(2):533–560, 2010.

[22] Laurens van der Maaten and Geoffrey Hinton. Visualizing data using t-sne. *Journal of Machine Learning Research*, 9(Nov):2579–2605, 2008.

[43] Sam T Roweis and Lawrence K Saul. Nonlinear dimensionality reduction by locally linear embedding. *Science*, 290(5500):2323–2326, 2000.

[33] Joshua B Tenenbaum, Vin De Silva, and John C Langford. A global geometric framework for nonlinear dimensionality reduction. *Science*, 290(5500):2319–2323, 2000.

[45] Sergei Sergeevich Vallender. Calculation of the wasserstein distance between probability distributions on the line. *Theory of Probability & Its Applications*, 18(4):784–786, 1974.

## Footnotes

[7]It is natural to consider $n - m$ dimensional volume for $f^{-1}(y)$, due to Sard's theorem [14] and implicit function theorem: since almost every $y \in f(B^n)$ is a regular value, $f^{-1}(y)$ is an $n-m$ dimensional submanifold, for such regular $y$. For an arbitrary continuous function, $Vol_{n-m} = \mathcal{M}_*^{n-m}$ is the lower Minkowski content, where the Waist Inequality is established [2]. For $n - m$ rectifiable sets, $Vol_{n-m} = \mathcal{M}_*^{n-m} = \mathcal{H}^{n-m}$.

[8]If the dimension of $f^{-1}(y)$ is greater than $n - m$, we define its volume to be $\infty$

[9]This is a weaker condition than Lipschitz, including functions of bounded variation. A Lipschitz function is differentiable almost everywhere.

[10]The Brenier theorem is not stated in the paper, but it holds under standard derivation.

[11] up to a measure zero set

[12] up to a measure zero set

[13] Such property can also be deduced from earlier work, e.g. Theorem 4.3 and Corollary 2.11 from [9]. But [18] is perhaps more direct and accessible.

[14]The regions $U$ and $f^{-1}(V)$ are given uniform distribution, i.e. their densities are $\frac{1}{Vol_n(U)}$ and $\frac{1}{Vol_n(f^{-1}(V))}$

[15]In our experiments we use POT: Python Optimal Transport library [12] to compute the Wasserstein distances.

[16] This issue was also discussed in [22].