[Reviews · NeurIPS 2018]

Reviewer 1



This paper investigates Dimensionality Reduction (DR) maps in an information retrieval setting. In particular, they showed that no DR map can attain both perfect precision and perfect recall. Further, they showed the theoretical bounds for the precision and the Wasserstein distance of a continuous DR map. They also run simulations in various settings. Quality: They have theoretical equivalences of precision and recall (Proposition 1) and show that perfect map does not exist (Theorem 1). They have theoretical upper bounds for precision in the worst case (Theorem 2) and average case (Theorem 3). They also have lower bounds for Wasserstein distance (Theorem 4, 5, 6). Theorems stated in this paper are sounds as far as I checked. Clarity: This paper is clearly written and well organized. Introduction gives motivation for why Dimension Reduction map is of interest, and Section 3 also gives motivation for why Wasserstein measure is in need. Originality: As the authors mentioned and also as I know, most literature on Dimensionality Reduction maps is heuristic without theoretical guarantee. Hence, as I know, this paper provides first theoretically sound results in DR maps. Significance: As in Originality, this paper poses important theoretical results in Dimensionality Reduction maps. Their proofs in the supplementary are also not trivial and important progress. Hence I would assess this paper as contributing to enhancement in dimensionality reduction. I have some suggestions in submission: 81th line: related problems in ?? -> related problems in Section 4 88th line: \omega^f (x) = inf_{U} -> \omega^f (x) = inf_{U:open}: i.e. infimum should be taken over open sets ------------------------------------------------------------------------------- Comments after Authors' Feedback I agree with other reviewers that Simulations section can be expanded and I would like to see that in Appendix Section, as in the authors' feedback. But I still appreciate their careful theoretical analysis on the problem setting (Precision and Recall) and Wasserstein bounds, so I would maintain my score. Additional comment: I noticed that you used vector images on p.24-26 in the supplementary, which makes the pdf file slower to view and print since pdf file is drawing thousands of points every time. I would suggest to change these to image files (e.g., png format) and embed those in the supplementary instead.

Reviewer 2



In this paper the authors study dimensional reduction (DR) maps from quantitative topology point of view and show that no DR map can achieve perfect precision and perfect recall simultaneously. The authors prove upper bound on precision of Lipschitz continuous DR map. To further quantify the distortion, the authors proposed a new measure based on L2-Wasseman distance and also provided lower bound for Lipschitz continue DR maps. Comments: 1. The paper advances theoretical understanding precision and recall when dimension reduction occurs. It has been observed in numerical experiments that there exists a tradeoff between precision and recall when dimensionality reduction happens, which was not previously well understood. This papers sheds some light towards that end. Main contribution of this paper is its technical proofs which this reviewer was unable to check in its entirety due to lack of time and expertise.

Reviewer 3



Paper Summary: The authors quantify the quality of a dimensionality reduction map in terms of its precision and recall. Taking a topological point of view, they derive several interesting results such as for sufficiently low dimensional mappings, perfect precision and recall is not possible simultaneously (Thm. 1, Prop. 2). They further quantify the tradeoff showing that for maps with perfect recall, precision rate decays exponentially (as a function of the ‘gap’ between the intrinsic and the embedding dimension of the input data) (Thm. 2, 3). They further refine their quality estimate from simple precision-recall tradeoff to essentially how bad are the recall elements in terms of their (Wasserstein) distance. They provide the analogous tradeoff bound for Wasserstein distance (Thm. 4). Review: * I want to disclose to the authors that I have reviewed a previous version of this manuscript * I really appreciate that the authors have significantly improved their manuscript and in my view merit a publication. Overall I really like the paper. The authors provide a nice theoretical perspective of what to expect from (sufficiently low dimensional) mappings from an information retrieval (precision/recall-type) perspective. This has significant implications in regards to what kinds of embedding results can one expect from popular visualization techniques such as t-SNE. This paper proves that any such embedding method cannot achieve both precision and recall simultaneously and therefore in a sense the practitioner should be careful in interpreting the embedding results from such visualizations. While I have not checked the proofs is detail (due to short review time and the appendix is 17 pages long), I do believe that the proof ideas are correct. I have a few minor suggestions which I really hope the authors would include in the final version of their manuscript. - As mentioned in my previous review, in the context of manifolds, analyzing the case of m < n < N (where m is the embedding dimension, n is the manifold dimension and N is the ambient dimension) feels very awkward to the reader. After all you are destroying the injectivity structure. While this difference is now discussed in Sec. 4, it would be nice to include a short discussion on merits of studying such maps in Sec. 1.1. This could be nicely motivated it by giving examples of visualization maps like t-SNE, and would overall strengthen the message of the paper. - There are some recent theoretical results on the cluster structures that t-SNE reveals. See recent work by Arora et al. “An Analysis of the t-SNE Algorithm for Data Visualization” in COLT 2018. I believe that a map that recovers correct clusters can be viewed as the one which has a specific precision/recall tradeoff. It would be nice to place Arora’s result for t-SNE in this paper’s context. I believe including a short discussion on this in Sec 4 would make the discussion more “up-to-date”. - One main criticism I have about the current work that the authors don’t explore or emphasize the that practical significance of their results. It would be nice to include a more detailed experiments section where they study (sufficiently) low dimensional mappings and explore some of the precision/recall tradeoffs. Having a detailed experimental section will significantly strengthen the overall message and applicability and significance of this paper. Minor comments/typos: - [line 80] “??” needs to be fixed. - [lines 136-137] “perfection” should be “perfect” - [Eq. 1] For completeness, define Gamma function. - [line 182] probabilities q1 and q2 are basically unreadable (because of the domain of integration), consider rephrasing it more aesthetically. - [line 197] “Gamma(P_U, P_f^-1(V))” Consider using a different symbol than Gamma (as it is already used in Eq. 1). - [line 234] “we then project these 10 dimensional…”, exactly what projection map is being used? Linear? Non-linear? - [line 282-283] while the C^infy case of Nash embedding is for a quadratic polynomial, the result in [28] is a (smooth) approximation to the C^1 Nash embedding result where the polynomial is in fact linear. ---- I have read the authors' response.

Reviewer 4



he authors proposed to relate the notion of continuity and precision/recall trade off, and derive upper bounds on precision over reduced dimensionality. The authors also propose the use of Wasserstein distance as a measure of distortion of distances in the original set and retrieved set. It is quite meaningful to explicitly show the relationship between precision, recall, and dimension reduction. The proposed evaluation of f by Wasserstein measure between uniform distribution on U, f^{-1}(V) is interesting. As far as I understand, the derivations are theoretically valid. I have positive impression for this paper, but presentation of the paper should be improved. For example, Fig.1 is uninformative. It is a pretty good explanation of precision and recall, but it is easily understandable from definition of precision and recall. Instead, it would be better to put graphical and conceptual description of, for example, waist inequality or isoperimetric inequality in theorem 6, which would be more informative and helpful for better understanding of the established results. I'm a bit confused the authors minimizes the lower bound of Wasserstein measure. Is it not very clear by this we can expect a good retrieval results. The authors claim that the experimental results are well aligned to those of optimized f-scores, but there should be more discussion on the reasoning to validate this evaluation. Also, figure 2 is too small and I cannot comprehend anything meaningful from the figure. Minor points: - between lines 67 and 68, \Gamma(,) is not defined yet. It is defined line 197. - line 81: unresolved reference. - line 90: d( x, A), namely, distance between a point and a set should be properly defined.